# Structural basis of regulated N-glycosylation at the secretory translocon

Melvin Yamsek[1,7], Mengxiao Ma[2,3,7], Roshan Jha[1,7], Yu Wan[4], Qianru Li[5], Frank Zhong[6], Katherine DeLong[2,3], Zhe Ji[4,5], Rajat Rohatgi[2,3✉] & Robert J. Keenan[1✉]

Most human secretory pathway proteins are N-glycosylated by oligosaccharyltransferase (OST) complexes as they enter the endoplasmic reticulum (ER)[1–3]. Recent work revealed a substrate-assisted mechanism by which N-glycosylation of the chaperone glucose-regulated protein 94 (GRP94) is regulated to control cell surface receptor signalling[4]. Here we report the structure of a natively isolated GRP94 folding intermediate tethered to a specialized CCDC134-bound translocon. Together with functional analysis, the data reveal how a conserved N-terminal extension in GRP94 inhibits OST-A and how structural rearrangements within the translocon shield the tethered nascent chain from inappropriate OST-B glycosylation. These interactions depend on a hydrophobic CCDC134 groove, which recognizes a non-native conformation of nascent GRP94. Our results define a mechanism of regulated N-glycosylation and illustrate how the nascent chain remodels the translocon to facilitate its own biogenesis.

Protein N-glycosylation is catalysed by OST complexes in the ER membrane. About 20% of human proteins are modified with N-glycans, which affect their folding, degradation, trafficking and function[1,2]. Defects in the cellular N-glycosylation machinery cause a group of inherited diseases called congenital disorders of glycosylation (CDG)[5], and pathogenic missense mutations resulting in a gain or loss of N-glycosylation have been described[6,7].

Metazoan genomes encode two OST complexes[3]. OST-A associates with ER membrane-bound ribosomes docked to SEC61 and TRAP (the 'secretory' translocon), giving it priority access to the unfolded nascent chain as it enters the lumen[8,9]. Sites skipped by OST-A can be modified by OST-B, which can act co- or post-translationally but does not associate with the ribosome–translocon complex (RTC)[10]. Notably, the presence of an acceptor sequon (Asn-X-Thr/Ser, X≠Pro) does not ensure its modification[11,12]. Factors such as sequence context, folding kinetics and cellular metabolic state influence acceptor site usage, leading to variability across cells, tissues and disease states[13,14]. The functional consequences of this variability remain poorly understood.

Glycosylation of normally unused acceptor sequons can target proteins for degradation by the ERAD pathway[15,16]. An important example is GRP94 (or HSP90B1), an abundant ER chaperone with six acceptor sequons (Fig. 1a). Whereas GRP94 is typically glycosylated at a single site (N217), glycosylation of up to five 'facultative' sites has been observed in cells subjected to different forms of ER stress[17–20].

A pathway regulating GRP94 hyperglycosylation was recently identified through genome-wide CRISPR–Cas9 screening[4]. This process depends critically on the lumenal ER protein CCDC134 and components of OST-A that recognize a conserved, metazoan-specific N-terminal extension of GRP94 to suppress glycosylation of its facultative sites during synthesis. Disrupting this pathway causes GRP94 hyperglycosylation and degradation and impairs the folding, maturation and

trafficking of its receptor clients (for example, IGF1R, LRP6 and TLR4) with broad effects on cell and tissue function[4,21,22]. Indeed, mutations that cause loss of CCDC134 function impair WNT and IGF1R signalling, causing the bone disorder osteogenesis imperfecta[4,23–25]. The structural basis by which these factors control GRP94 acceptor site usage at the secretory translocon remains unclear.

## CCDC134 and FKBP11 recruitment

In earlier work, we used affinity purification, mRNA sequencing and mass spectrometry to identify FKBP11 as a transmembrane protein that can bind ribosome-associated secretory translocons during synthesis of proteins with long lumenal segments[26]. Notably, CCDC134 and GRP94 were identified as co-purifying factors. In a reciprocal experiment, FKBP11 was recovered following affinity purification by means of a FLAG tag on CCDC134, and this interaction was dependent on the ribosome, as neither FKBP11 nor CCDC134 recovered the other component in the absence of ribosomes (Extended Data Fig. 1a,b).

To identify cotranslational clients of FKBP11 and CCDC134, we used a recently developed ribosome profiling method optimized for low-input samples[27,28] (Extended Data Fig. 1c–e). As observed previously for FKBP11 (ref. 26), affinity-purified CCDC134-bound ribosomes were modestly enriched for transcripts encoding proteins with long translocated segments (Fig. 1b). This seemed to depend on FKBP11, as most transcripts enriched by CCDC134 were also enriched by FKBP11, but not vice versa (Fig. 1c). Notably, GRP94 was strongly enriched by both factors and was also the most abundant transcript in each elution (Fig. 1c–e).

Positional analysis revealed distinct interaction profiles for FKBP11 and CCDC134 during synthesis of the 803-residue GRP94 polypeptide. FKBP11 arrived first, when the nascent chain was about 161 residues

[1]Department of Biochemistry and Molecular Biology, The University of Chicago, Chicago, IL, USA. [2]Department of Biochemistry, Stanford University School of Medicine, Stanford, CA, USA. [3]Department of Medicine, Stanford University School of Medicine, Stanford, CA, USA. [4]Department of Biomedical Engineering, McCormick School of Engineering, Northwestern University, Evanston, IL, USA. [5]Department of Pharmacology, Feinberg School of Medicine, Northwestern University, Chicago, IL, USA. [6]Department of Molecular Genetics and Cell Biology, The University of Chicago, Chicago, IL, USA. [7]These authors contributed equally: Melvin Yamsek, Mengxiao Ma, Roshan Jha. ✉e-mail: rrohatgi@stanford.edu; bkeenan@uchicago.edu

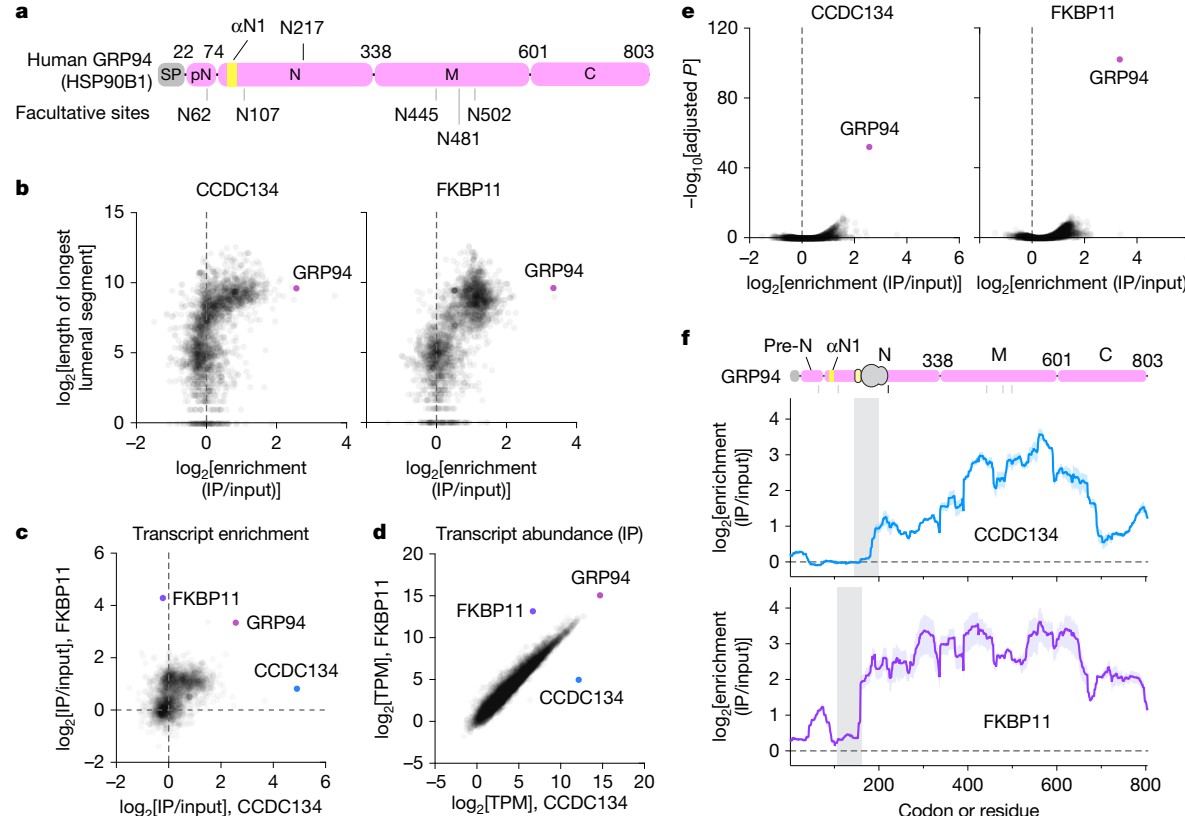

**Fig. 1 | CCDC134 and FKBP11 are stabilized at the translocon during GRP94 synthesis. a**, Domain organization of GRP94, highlighting the one constitutive (N217) and five facultative N-glycosylation acceptor sites. The signal peptide (SP), pre-N segment (pN) and first helix of the N-domain (αN1, yellow) are also indicated. **b**, $\log_2$[transcript enrichment] (IP abundance/input abundance) plotted against $\log_2$[length of the longest lumenal (translocated) segment]; data are the average of two biological replicates for all secretory and transmembrane transcripts recovered following FLAG–CCDC134 or FLAG–FKBP11 affinity purification ($n$ = 2,296). **c**, Comparison of $\log_2$[transcript enrichment] for all secretory and transmembrane transcripts recovered in the CCDC134 and FKBP11 samples. Note that FKBP11 and CCDC134 are among the most enriched transcripts in their respective samples, because each protein is tagged at its N terminus (immediately following the SP); this serves as an internal control for the immunoprecipitation. **d**, Comparison of $\log_2$[transcript abundance] (transcripts per million, TPM) for all transcripts recovered following FKBP11 and CCDC134 affinity purification ($n$ = 9,242). **e**, Volcano plots for all secretory and transmembrane transcripts showing $-\log_{10}$[adjusted $P$-value] versus $\log_2$[enrichment] for each sample. For clarity, FKBP11 and CCDC134 transcripts are not shown. **f**, CCDC134 and FKBP11 interaction profiles for GRP94; mean (line) and range (shading) for two biological replicates are indicated. GRP94 residues to the left of the vertical grey bars are in the lumen at the onset of CCDC134 and FKBP11 engagement, respectively. Diagram of GRP94 and the position of the ribosome (grey) and SEC61 channel (yellow) at the onset of CCDC134 engagement are shown at the top. For source data related to panels **b**–**e**, see Supplementary Table 1.

long (Fig. 1f). At this point, GRP94 residues 22–106 were in the lumen (assuming that approximately 55 residues are buried in the ribosome exit tunnel and SEC61 channel). CCDC134 began to arrive shortly thereafter, when the nascent chain was about 204 residues long. At this point, GRP94 residues 22–149 were in the lumen, including all of the pre-N segment (residues 22–74) and the first helix of the N-domain (αN1, residues 82–91). This is consistent with a previous analysis of GRP94 translocation intermediates assembled by in vitro translation[4]. CCDC134 was stabilized further as synthesis continued, peaking as the GRP94 N-domain finished translocating into the lumen. Once recruited, both factors persisted until translation was terminated. Thus, FKBP11 and CCDC134 engage the translocon early during GRP94 synthesis and remain bound until synthesis is complete.

## Structure of translocon-bound GRP94

To understand how these factors are organized, we prepared FKBP11-bound ribosomes from FLAG–FKBP11 HEK293 cells and solved the structure by single-particle cryo-electron microscopy (cryo-EM) (Fig. 2a, Extended Data Figs. 2–6 and Extended Data Table 1). The maps show well-defined density for SEC61, OST-A, A/P- and P/E-site transfer RNAs and the ribosome. Density was also visible for the TRAP complex, portions of the lipid-linked oligosaccharide (LLO) donor and several N-glycans. Nascent chain density was visible in the ribosome exit tunnel and SEC61 channel, which adopts an open conformation with its displaced plug helix packed against SEC61β, SEC61γ and DC2 (ref. 29), and RAMP4 bound at the lateral gate[30]. Thus, these natively isolated FBKP11–ribosome complexes represent physiologically relevant complexes captured in the act of translation and translocation.

Additional lumenal density was visible near OST-A (Fig. 2a). This density is distinct from weak, unassigned densities visible near OST-A in a recent cryo-electron tomography (cryo-ET) study[29]. Although FKBP11 and CCDC134 do not interact in the absence of ribosomes, AlphaFold predicted a high-confidence complex that was fit into a portion of this density with only minor adjustments to each subunit's flexible C-terminal region (Fig. 2b and Extended Data Figs. 4a and 5a). This configuration buries the canonical ligand binding site of FKBP11 and positions its conserved and positively charged C-terminal helix, previously shown to be important for ribosome binding[26], adjacent to uL22 and 28S rRNA H24 on the cytosolic side of the membrane (Extended Data Fig. 6f,g).

The remaining density was assigned to residues 25–283 of nascent GRP94, which is observed in a non-native conformation (Fig. 2c). The pre-N segment (residues 22–74), which is largely unstructured or disordered in previously determined structures of GRP94, was modelled

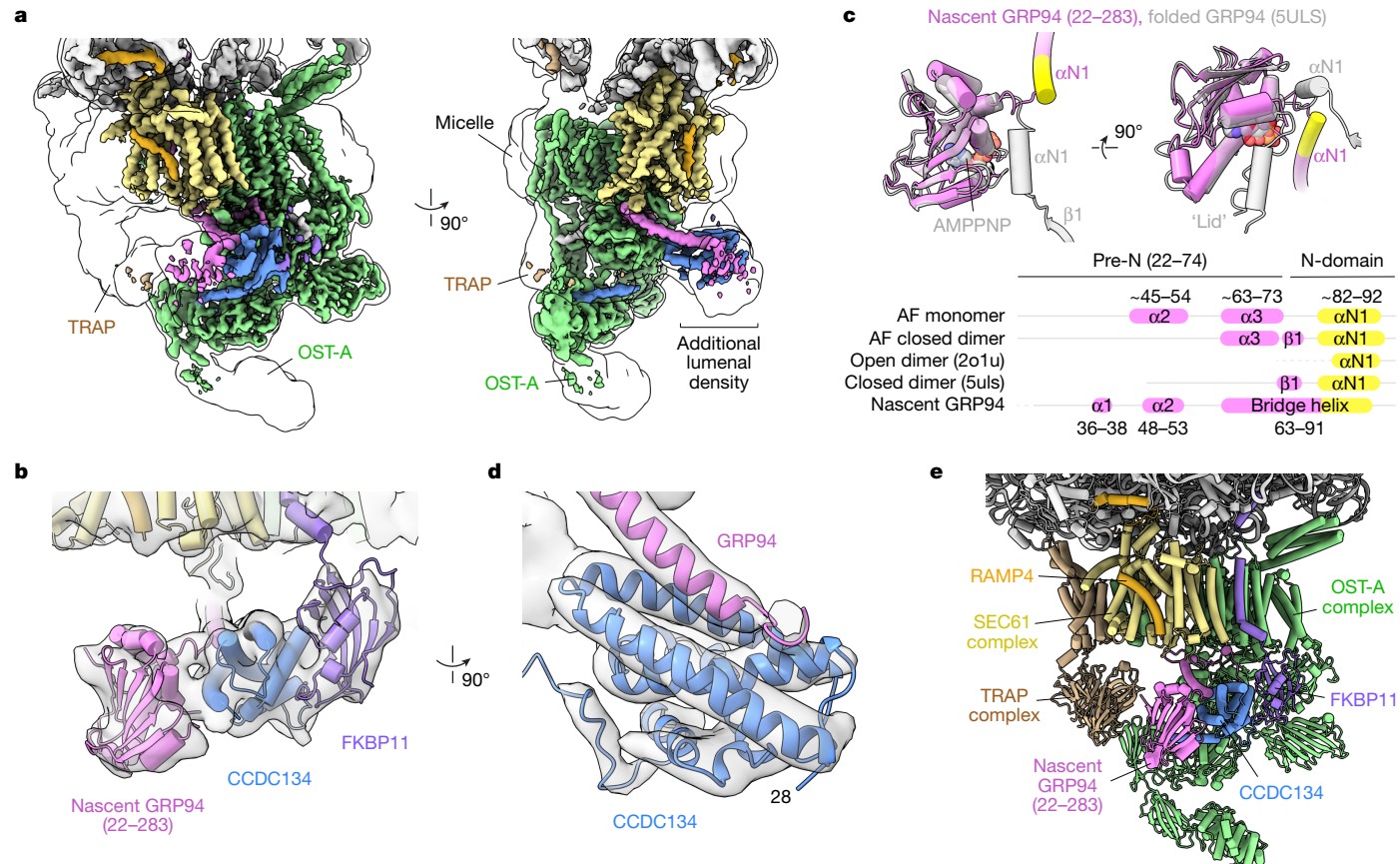

**Fig. 2 | Structure of nascent GRP94 tethered to a CCDC134- and FKBP11-bound secretory translocon. a**, Overview of the cryo-EM translocon map after post-processing using DeepEMhancer. The outline shows the same map low-pass filtered to 10 Å resolution. **b**, Close-up of CCDC134 (blue), FKBP11 (purple) and nascent GRP94 (magenta) fit to the cryo-EM translocon map low-pass filtered to 8 Å resolution. **c**, Structural comparison of folded, nucleotide-bound GRP94 (PDB ID 5ULS; grey) and nascent GRP94 (magenta), showing conformational differences in the lid and helix αN1 (yellow). Secondary structure annotation for the GRP94 pre-N segment and flanking regions is shown at the bottom, based on experimental structures and AlphaFold (AF) models. **d**, Close-up of CCDC134 and part of the GRP94 'bridge' helix fit to the cryo-EM translocon map low-pass-filtered to 6 Å resolution. **e**, Overview of the final model showing the arrangement of CCDC134, FKBP11 and nascent GRP94 relative to components of the ribosome-bound secretory translocon.

into continuous density that wraps around the OST-A active site, in good agreement with high-confidence AlphaFold modelling (Extended Data Figs. 4b and 5b). AlphaFold was also used to guide placement of the first N-domain helix (αN1, residues 82–91), which normally packs against the ATP-binding site in folded GRP94 (Fig. 2c), into density extending along the surface of CCDC134 (Fig. 2d and Extended Data Figs. 4c and 5c). Portions of the globular N-domain, including its central helix and most of the eight-stranded β-sheet, were placed into weak but characteristic density adjacent to CCDC134 (Fig. 2b and Extended Data Fig. 6h). No density was observed for bound nucleotide, the flexible N-domain active-site 'lid' or the M- and C-terminal domains. Thus, the structure reveals a bona fide GRP94 translation intermediate(s) captured in a partially folded state while bound to the secretory translocon and two accessory factors, FKBP11 and CCDC134 (Fig. 2e).

### Nascent GRP94 inhibits OST-A

The pre-N segment extends deep into the OST-A active-site cavity, burying more than 3,100 Å$^2$ of surface area against STT3A (about 1,680 Å$^2$), DC2 (about 540 Å$^2$) and SEC61α (about 930 Å$^2$) (Fig. 3a,b). This negatively charged region (residues 25–79) (Fig. 3c,d) is rich in conserved residues[4] and forms an extensive network of intra- and inter-molecular contacts. This region is highly sensitive to mutation, as most variants tested in RKO cells led to GRP94 hyperglycosylation and degradation and, consequently, to reduced cell surface expression of its clients

IGF1Rβ and LRP6 (Fig. 3e). Notably, these mutations did not affect GRP94 folding or chaperone activity, as they did not impair the activity of a variant lacking all five facultative sequons (5N; Extended Data Fig. 7b). Thus, the metazoan-specific pre-N segment regulates GRP94 hyperglycosylation, not its catalytic activity.

The pre-N interactions within the acceptor site binding pocket mimic those of previously determined structures of short (about seven residues) peptides bound to yeast OST and human OST-B (refs. 31,32) (Fig. 3f). Density was observed for portions of the LLO glycan, polyprenyl chain and pyrophosphate, but not an active-site divalent cation (Extended Data Fig. 6b). The lumenal EL5 loop of STT3A is in an ordered, 'engaged' conformation, packed against the LLO and GRP94. Notably, the conserved GRP94 'SRT' motif (residues 42–44) forms a tight 180° turn that positions S42 close to the donor LLO substrate (Fig. 3g). Here, the hydroxyl group of T44 contacts the STT3A WWD motif, which is essential for acceptor sequon recognition[33,34], explaining why mutations in either site lead to GRP94 hyperglycosylation and degradation[4].

The central (+1) position of an acceptor sequon can be any residue except proline, which is incompatible with the tight turn conformation required for OST-A binding. As expected of a substrate mimic, an arginine-to-proline mutation at GRP94 position 43 resulted in hyperglycosylation and degradation (Fig. 3e). Notably, mutating this position to a negatively charged glutamate (and, to a lesser extent, alanine or isoleucine) also caused GRP94 hyperglycosylation, whereas lysine was tolerated[4]. A positive charge at the +1 position of GRP94 is probably

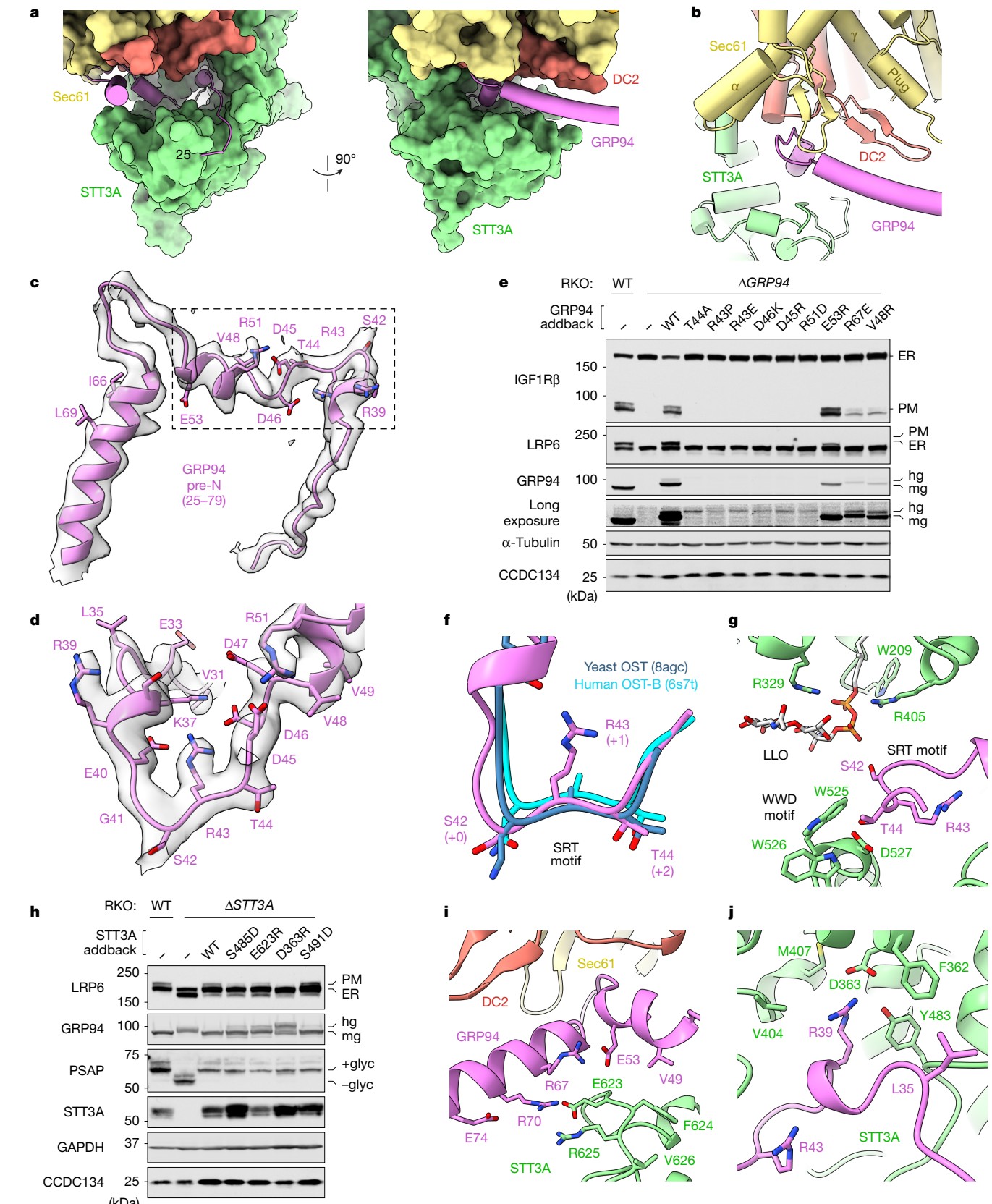

**Fig. 3 | See next page for caption.**

required to neutralize the flanking negative charges on E40 and D46 that are brought together on binding (Fig. 3d).

These observations define the pre-N segment as a pseudo-substrate inhibitor that occludes the STT3A acceptor site binding pocket during GRP94 synthesis to limit glycosylation of the facultative sites. This mechanism differs from that proposed for the small-molecule OST inhibitor NGI-1, which traps a catalytically inactive conformation of LLO but does not prevent acceptor peptide binding[21].

**Fig. 3 | The nascent GRP94 pre-N segment is a pseudo-substrate inhibitor of OST-A. a**, Orthogonal views of the nascent GRP94 pre-N segment (magenta) wedged between two OST-A subunits, STT3A (green) and DC2 (salmon), and SEC61α (yellow). **b**, Close-up showing pre-N segment contacts with β-hairpins in DC2 and SEC61α. The translocating GRP94 chain displaces the SEC61α plug, which packs against the DC2 β-hairpin and SEC61γ. **c**, Overview of the pre-N segment fit to the cryo-EM translocon map after low-pass filtering by local resolution. The positions of residues tested in cells are indicated. **d**, Close-up of the same cryo-EM map, highlighting residues in the pre-N SRT region. **e**, ER and plasma membrane (PM) levels of IGFR1β and LRP6 and levels of monoglycosylated (mg) and hyperglycosylated (hg) GRP94 in *GRP94* knockout RKO cells stably expressing wild-type (WT) GRP94 or the indicated variant. **f**, Superposition of OST structures bound to short substrate analogues (cyan, blue) or the SRT motif of nascent GRP94 (magenta). **g**, Organization of the GRP94 SRT motif and LLO within the STT3A (green) active site. **h**, Glycosylation of GRP94 and PSAP (an OST-A specific substrate) and cell surface abundance of LRP6 in *STT3A* knockout RKO cells stably expressing WT STT3A or the indicated variant. Close-up views showing the location of E623R (**i**) and D363R point mutants in STT3A (green) (**j**) that selectively affect GRP94 (but not PSAP) glycosylation. Data shown in panels **e** and **h** are representative of at least two independent experiments. For gel source data, see Supplementary Fig. 1.

The persistent binding of the 53-residue pre-N segment contrasts with the transient engagement of nascent polypeptides as they enter the lumen, suggesting differences in how they interact with OST-A[35–37]. Unlike the conserved and extensive interface used by the pre-N domain, recognition of bona fide acceptor sequences is primarily mediated by contacts between the acceptor sequon and STT3A[31,32,38]. Consistent with this distinction, we identified distal STT3A variants (D363R and E623R) that caused GRP94 hyperglycosylation and reduced surface expression of LRP6 but had no effect on the OST-A-specific substrate prosaposin (PSAP)[10] (Fig. 3h–j). Notably, these residues are close to the known CDG mutations Y360S and V626A[39–41], the latter of which causes GRP94 hyperglycosylation in patient fibroblasts[19]. This suggests that the disease phenotypes of some CDG-causing *STT3A* mutations may be because of GRP94 and its clients, rather than (or as well as) a general N-glycosylation defect.

## CCDC134 stabilizes nascent GRP94

Pre-N binding to OST-A is necessary but not sufficient to fully suppress GRP94 hyperglycosylation, which also requires CCDC134 (ref. 4). CCDC134 is recruited by a network of interactions with the translocon and nascent GRP94 (Fig. 4a). More than 2,100 Å² of CCDC134 surface area is buried in contacts with FKBP11 (about 790 Å²) and STT3A (about 320 Å²) through its globular domain, and OST48 (about 630 Å²), RPN2 (about 330 Å²) and RPN1 (about 110 Å²) through its long C-terminal extension. Although FKBP11 is well positioned to help recruit CCDC134, this is not strictly required, because HEK293 and RKO cells lacking FKBP11 did not show a GRP94 hyperglycosylation defect (Extended Data Fig. 7c,e). However, STT3A variants carrying mutations in the small CCDC134–STT3A interface were unable to fully suppress GRP94 hyperglycosylation[4]. Similarly, a CCDC134 variant lacking its OST-A-binding C-terminal extension (Δ194–224) (but retaining the 'QSEL' ER retention signal) did not fully rescue a *CCDC134* knockout, probably because of impaired assembly at the translocon (Fig. 4e and Extended Data Fig. 7f).

The globular domain of CCDC134 makes further interactions with nascent GRP94, burying almost 1,100 Å² of CCDC134 surface area in two distinct interfaces (Fig. 4b). The smaller interface (about 350 Å²) involves the partially folded GRP94 N-domain, which packs loosely against CCDC134. The larger interface (about 720 Å²) involves the amphipathic GRP94 helix αN1 (including residues 82–91 and several flanking residues), which flips about 180° from its canonical position in folded GRP94 to pack against a conserved hydrophobic groove on the surface of CCDC134 (Fig. 4b,c). In this configuration, the canonical αN1 binding site in the N-domain is occluded by CCDC134 (Fig. 4b). Notably, the αN1 region is required for CCDC134-dependent suppression of GRP94 hyperglycosylation[4]. Consistent with this, CCDC134 variants carrying structure-guided hydrophobic-to-charge mutations within the hydrophobic groove were recruited to the translocon but did not prevent GRP94 hyperglycosylation in *CCDC134* knockout RKO cells (Fig. 4d,e and Extended Data Fig. 7f). Thus, CCDC134 binding to αN1 and pre-N binding to the OST-A active site are necessary to fully suppress GRP94 hyperglycosylation.

In nascent GRP94, αN1 and the pre-N segment are connected by a long helical element, termed the bridge helix (residues 63–91) (Fig. 4a,b). This helix is not observed in experimental and predicted structures of GRP94, which show considerable conformational flexibility in this region[42,43] (Fig. 2c). The paucity of direct contacts between CCDC134 and STT3A suggested that rigidity of the bridge helix may be functionally important. Consistent with this, individual helix-disrupting proline substitutions at positions along the bridge helix (I66, L69) caused GRP94 hyperglycosylation, whereas alanine substitutions were tolerated (Fig. 4f and Extended Data Fig. 7g). Notably, these mutations did not affect folding or chaperone activity of the mature protein, as they did not impair the activity of the 5N variant. Thus, the bridge helix promotes GRP94 binding by rigidifying the connection between the two key regulatory surfaces of CCDC134 and OST-A.

These data show how CCDC134 recognizes a non-native conformation of nascent GRP94 to occlude the OST-A active site and tether the growing polypeptide chain to the translocon. This explains why CCDC134 is stabilized at the secretory translocon during GRP94 synthesis[4] (Fig. 1f) and why fully folded GRP94 does not inhibit OST-A or OST-B post-translationally, despite its abundance in the ER lumen.

## A lumenal vestibule shields GRP94 from OST-B

The pre-N and N-domain of GRP94 remain tethered to the translocon until translation is terminated (Fig. 1f). Density for the M- and C-domains—which account for nearly 60% of the mature protein—is not visible in the structure, and superimposed full-length GRP94 monomers[42,43] show numerous intra- and inter-molecular clashes, including with CCDC134 and subunits of OST-A (Extended Data Fig. 8). This suggests that the native N–M and M–C domain interfaces cannot form until synthesis is complete.

The TRAP complex adopts different positions in previously determined translocon structures[29,44,45]. In the CCDC134-bound secretory translocon, it shifts away from the tethered and partially folded GRP94 N-domain. Relative to a previously determined cryo-ET structure of the secretory translocon, the TRAP transmembrane bundle moves away from the SEC61γ N-terminus and the TRAPα TMD is no longer visible near the SEC61α hinge[29] (Fig. 5a). In its place, we observed density for a small four-helix bundle, which was assigned to KCP2, the DC2-binding subunit of OST-A (refs. 9,46–48) (Fig. 5a and Extended Data Figs. 4d and 5d). The DC2 N-terminus extends more than 20 Å along the membrane to bind KCP2 by means of a cleft on its cytosolic face. This interaction buries more than 1,200 Å² of DC2 surface area and loosely tethers KCP2 near the SEC61α hinge, SEC61γ, the ribosome and the repositioned TRAP TMDs.

Notably, the TRAP lumenal domain shifts from its position at the SEC61α hinge loop[44] to form a new interface with RPN2 (Fig. 5b,c). This creates a large lumenal vestibule adjacent to the SEC61 channel exit that is lined by TRAP, OST-A, CCDC134 and the partially folded GRP94 N-domain (Fig. 5d). Here, the downstream domains of tethered GRP94 can begin folding in close proximity to the translocon, thereby limiting inappropriate cotranslational glycosylation of the M-domain by OST-B acting in *trans*[10].

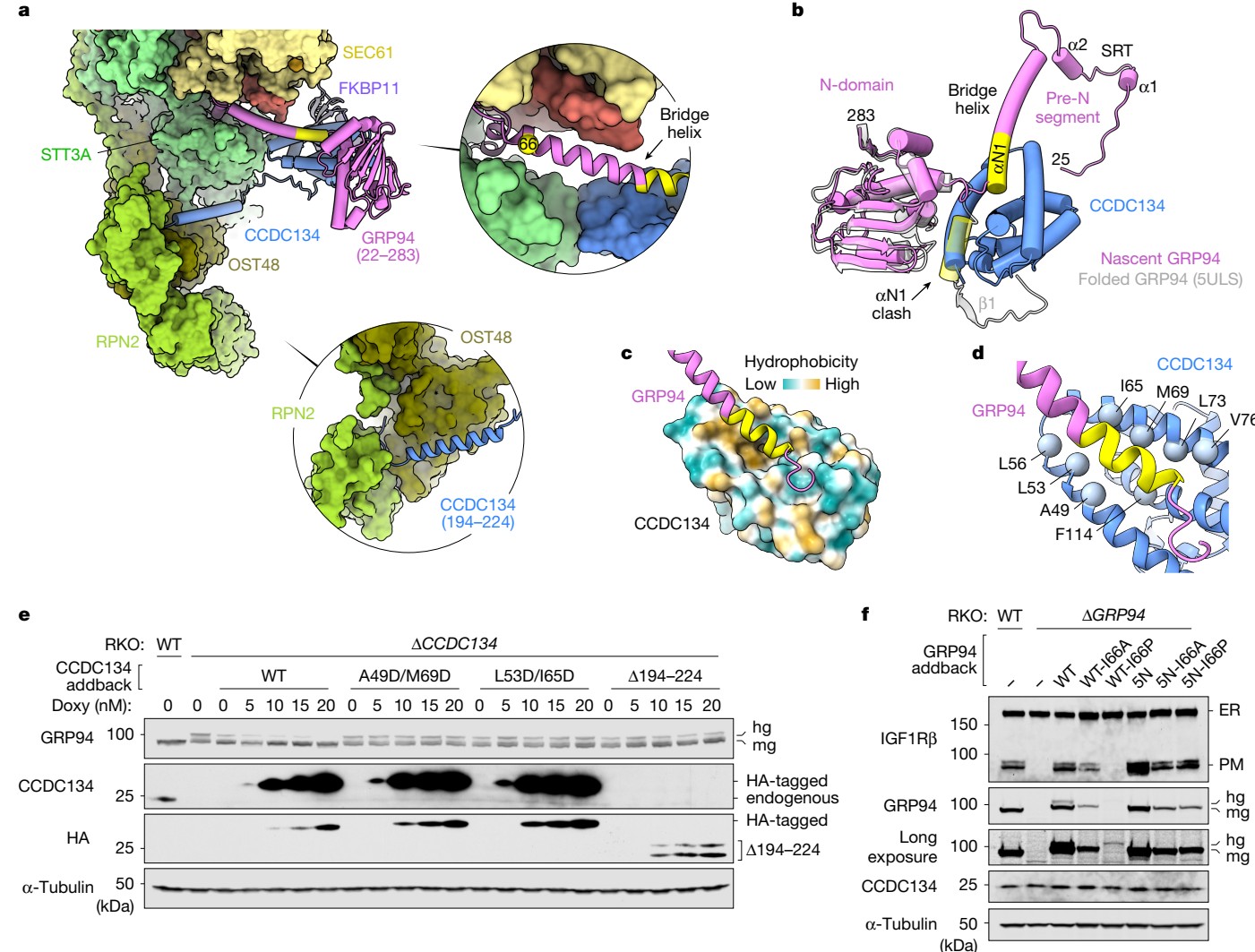

**Fig. 4 | A hydrophobic groove in CCDC134 stabilizes a non-native conformation of nascent GRP94. a**, Overview of CCDC134 (blue) interactions at the secretory translocon. The top inset shows a close-up of the GRP94 'bridge' helix (magenta, with αN1 in yellow) connecting the OST-A active site (green) and CCDC134. The bottom inset shows the helical C-terminal extension of CCDC134 packed against a cleft between OST48 (olive) and RPN2 (lime green). **b**, Structural comparison of folded, nucleotide-bound GRP94 (PDB ID 5ULS; grey) and nascent GRP94 (magenta), showing CCDC134 bound to a non-native conformation of nascent GRP94. In this tethered conformation, the canonical αN1 binding site in the N-domain is occluded by CCDC134. **c**, GRP94 αN1 binding surface of CCDC134 coloured by hydrophobicity. **d**, Close-up of hydrophobic residues lining the CCDC134 groove. **e**, Glycosylation of GRP94 in CCDC134 knockout cells stably expressing wild-type (WT) CCDC134 or the indicated variant. The Δ194–224 construct is only visible in the anti-HA blot because the anti-CCDC134 antibody recognizes a C-terminal epitope that is deleted from this construct. **f**, ER and plasma membrane (PM) levels of IGFR1β and levels of monoglycosylated (mg) and hyperglycosylated (hg) GRP94 in *GRP94* knockout RKO cells stably expressing the indicated GRP94 or GRP94-5N variant. The location of I66 along the bridge helix is shown in panel **a**. See also Extended Data Fig. 7g. Data shown in panels **e** and **f** are representative of at least two independent experiments. For gel source data, see Supplementary Fig. 1.

## Discussion

Our findings lead to a molecular model for regulated GRP94 N-glycosylation (Fig. 5e). FKBP11 is recruited to the secretory translocon as the pre-N segment translocates through SEC61 and begins to occlude the OST-A active site. Once FKBP11 is bound, CCDC134 is progressively stabilized through a network of interactions with FKBP11, OST-A and the elongating GRP94 chain. Key to this is binding of the CCDC134 hydrophobic groove to a transiently exposed amphipathic helix (αN1) in GRP94. This configuration tethers GRP94 to the translocon and stabilizes pre-N binding to the STT3A active site to prevent glycosylation of downstream sites by OST-A. The relatively complex architecture and high contact order[49] of the N-domain suggest that it folds slowly, leaving its constitutive glycosylation site (N217) accessible to OST-B (acting in *trans*) as it enters the lumen. Once packed against CCDC134, however,

the partially folded N-domain contributes to a lumenal vestibule that shields downstream facultative sites from OST-B while the M-domain attempts to fold. As synthesis finishes and GRP94 is released, the αN1 helix packs against the N-domain, leaving the mature (folded) GRP94 incapable of inhibiting OST-A or OST-B.

This pathway does not seem to be strictly required for proper GRP94 folding, as constructs lacking all five facultative sites are functional in RKO cells, even when harbouring mutations that disrupt their interactions with OST-A (Extended Data Fig. 7b) or in cells lacking CCDC134 or STT3A[4]. This raises the question of why the facultative sites are conserved. We propose that they are part of a cotranslational quality control system that has evolved to rapidly detect and degrade GRP94 misfolding events. This would be analogous to cotranslational ubiquitination of nascent polypeptides on cytosolic ribosomes[50–52], which preferentially targets proteins that fold inefficiently. Notably, GRP94 is a large

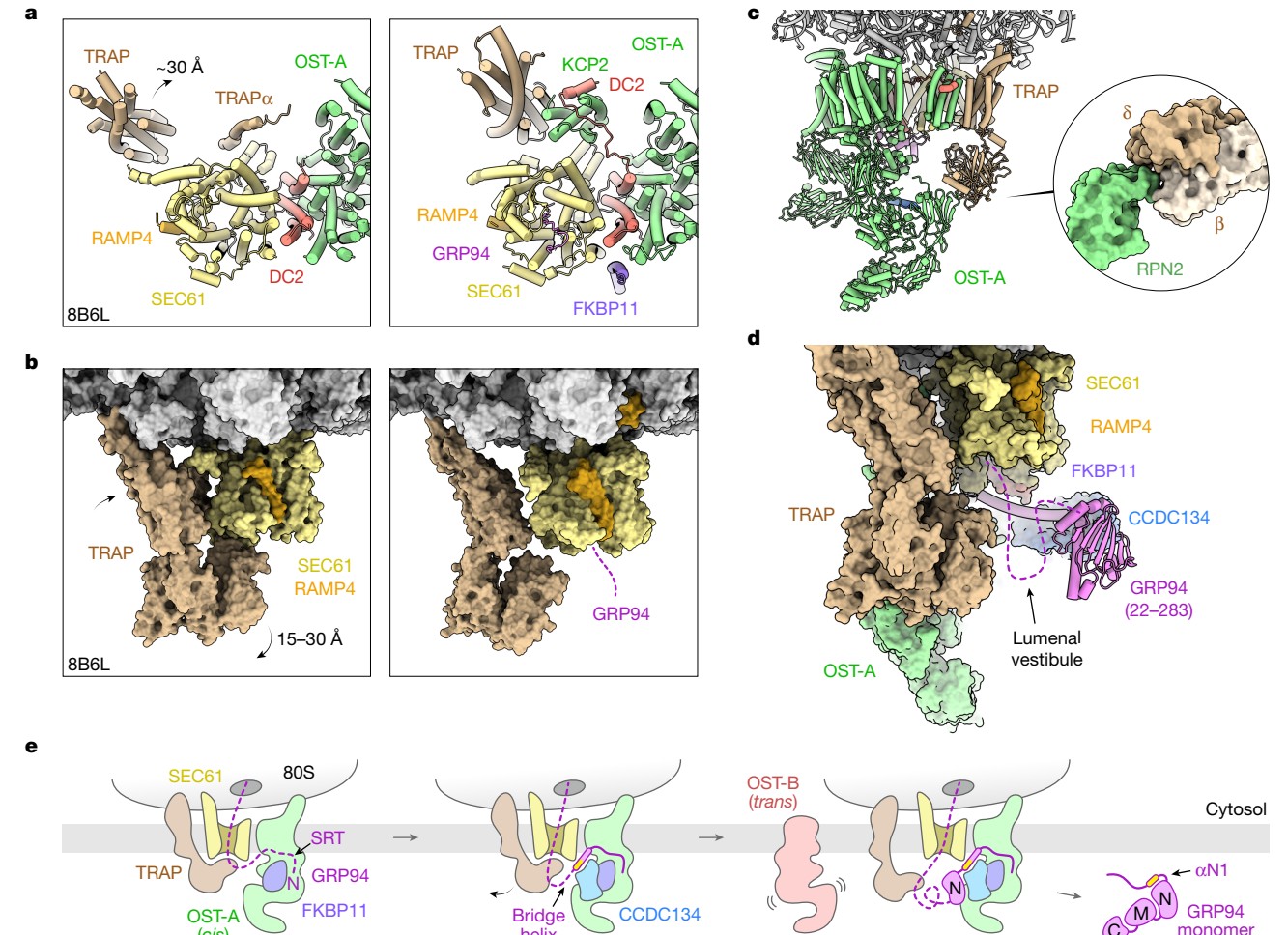

**Fig. 5 | A lumenal vestibule sequesters tethered GRP94 from OST-B.**
**a**, Comparison of subunit organization near the SEC61α hinge, viewed from the cytosol (left, cryo-ET structure of the secretory translocon, PDB ID 8B6L; right, present structure). Arrow indicates movement of the TRAP transmembrane segments, which is accompanied by displacement of the TRAPα TMD. The KCP2 subunit of OST-A is visible in its place, stabilized by a long N-terminal extension of DC2 (salmon) and packed loosely against the TRAP transmembrane bundle and the SEC61α hinge. **b**, Comparison of TRAP complex position in actively

translocating RTCs, highlighting the large 15–30-Å displacement of its lumenal domains. RAMP4 (orange) is bound at the lateral gate of fully open, plug-displaced SEC61 (yellow) in both structures. **c**, View of a new interface formed between RPN2 (green) and the repositioned TRAPδ (tan) and TRAPβ (light tan) lumenal domains (inset). **d**, View of the lumenal vestibule formed by TRAP, OST-A, CCDC134 and the nascent GRP94 N-domain. **e**, Model for regulated GRP94 N-glycosylation. For clarity, only the lumenal domain of FKBP11 is depicted. See text for details.

multidomain and oligomeric protein, prone to misfolding and aggregation[53,54]. It is also one of the most abundant proteins in the ER lumen. Thus, even a fraction of misfolded GRP94 might be sufficiently toxic in certain cell types or developmental stages to necessitate a dedicated quality control checkpoint at the RTC. Rather than facilitating folding, the function of CCDC134 would be to shield the facultative sites from glycosylation—by occluding the OST-A active site and restricting access of the tethered nascent chain to OST-B—while GRP94 attempts to fold.

Our data suggest that FKBP11 functions as a ribosome-binding adaptor to help recruit CCDC134 to the secretory translocon during GRP94 biosynthesis. Burial of its nominal catalytic site by CCDC134 argues against a role as a nascent chain prolyl isomerase. Although we did not detect GRP94 hyperglycosylation defects in knockout HEK293 or RKO cells, FKBP11 may be required in other contexts. Indeed, FKBP11 is highly expressed in specialized secretory cells and is upregulated in response to various ER stresses[55–58]. Notably, FKBP11 knockdown was recently reported to suppress proinflammatory cytokine production in endothelial cells by disrupting NF-κB signalling[59], consistent with a link to GRP94 biogenesis and cell-surface receptor signalling.

Our selective ribosome profiling data hint at an additional, more general function for FKBP11 and CCDC134 at the translocon. Hundreds of transcripts were enriched by FKBP11 and CCDC134, albeit at much lower levels than GRP94 (Fig. 1c). Notably, these proteins lack obviously conserved pre-N-like sequences, suggesting that FKBP11 and CCDC134 do not regulate their glycosylation. An intriguing possibility is that FKBP11 and CCDC134 also function more broadly to shield aggregation-prone hydrophobic segments that are transiently exposed during translocation into the ER lumen.

An important concept emerging from recent work is that the subunit composition of the translocon is dynamic[29,60–62]. Understanding how different substrates drive translocon remodelling to support their own biogenesis is an important goal. For nascent GRP94, a key signal is the transient exposure of an amphipathic helix that is recognized by CCDC134 to suppress glycosylation at the secretory translocon. Further studies are needed to explore how the cotranslational folding and modification of other nascent polypeptides is coordinated at the translocon.

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

# Methods

## Antibodies

The following primary antibodies were used: mouse anti-CCDC134 (Santa Cruz Biotechnology, #sc-393390, RRID: AB_3662100, 1:500); mouse anti-GRP94 (R&D Systems, #MAB7606, RRID: AB_3644153, 1:2,000); mouse anti-GRP94 (Santa Cruz Biotechnology, #sc-393402, RRID: AB_2892568, 1:2,000); rabbit anti-LRP6 (Cell Signaling Technology, #2560, RRID: AB_2139329, 1:1,000); mouse anti-α-tubulin (MilliporeSigma, #T6199, RRID: AB_477583, 1:10,000); mouse anti-α-tubulin (Abcam, #ab11304, RRID: AB_297909, 1:1,000); mouse anti-STT3A (Abnova, #H00003703-M02, RRID: AB_530104, 1:1,000); rabbit anti-STT3A (Proteintech, #12034-1-AP, RRID: AB_2877818, 1:1,000); rabbit anti-IGF1Rβ (Cell Signaling Technology, #9750, RRID: AB_10950969, 1:1,000); mouse anti-GAPDH (Proteintech, #60004-1-Ig, RRID: AB_2107436, 1:10,000); rabbit anti-PSAP (GeneTex, #GTX101064, RRID: AB_2037779, 1:1,000); mouse anti-HA (GenScript, #A01244, RRID: AB_1289306, 1:1,000); rabbit anti-HA (Bethyl, #A191-102, RRID: AB_2891412, 1:2,000); rabbit anti-uL22 (Abcepta, #AP9892b, RRID: AB_10613776, 1:1,000); rabbit anti-uL2 (Abcam, #ab169538, RRID: AB_2714187, 1:1,000); rabbit anti-SEC61β (Cell Signaling Technology, #14648, RRID: AB_2798555, 1:1,000); rabbit anti-TRAPα (MilliporeSigma, #HPA011276, RRID: AB_1857503, 1:1,000); rabbit anti-STT3B (Proteintech, #15323-1-AP, RRID: AB_2198046, 1:1,000); rabbit anti-FKBP11 (Atlas Antibodies, #HPA041709, RRID: AB_10794487, 1:1,000); mouse anti-FLAG (MilliporeSigma, #F1804, RRID: AB_262044, 1:1,000); mouse anti-BiP (BD Transduction Laboratories, #610978, RRID: AB_398291, 1:1000).

The following secondary antibodies conjugated to horseradish peroxidase were used: Peroxidase AffiniPure Donkey Anti-Mouse IgG (H+L) (Jackson ImmunoResearch Laboratories, #715-035-150, RRID: AB_2340770, 1:10,000); Peroxidase AffiniPure Donkey Anti-Rabbit IgG (H+L) (Jackson ImmunoResearch Laboratories, #111-035-144, RRID: AB_2307391, 1:10,000); Peroxidase Donkey Anti-Rabbit IgG (Fc specific) (Sigma-Aldrich, #SAB3700863, RRID: AB_3675584, 1:10,000); Peroxidase Rabbit Anti-Mouse IgG H&L (Abcam, #ab6728, RRID: AB_955440, 1:10,000). Secondary antibodies conjugated to IRDye 800CW were obtained from LI-COR (IRDye 800CW Donkey anti-Mouse IgG Secondary Antibody, 1:10,000).

## Constructs

**CCDC134 constructs.** Doxycycline-inducible 3xFLAG–CCDC134 (tag inserted after the ER signal sequence) was cloned by polymerase chain reaction (PCR) amplification followed by Gibson assembly (New England Biolabs) into pcDNA5-FRT/TO. Doxycycline-inducible 3xHA–CCDC134 (tag inserted after the ER signal sequence) mutants were cloned by PCR amplification followed by Gibson assembly (New England Biolabs) into pLenti-TRE-rtta3G-BLAST[4].

**GRP94 constructs.** pENTR2B-3xFLAG-GRP94 (wild type) and pENTR2B-3xFLAG-GRP94[5N] (includes mutations in all five facultative sequons: S64A, S109A, S447A, T483I, T504I)[4] were used as templates to clone all mutant constructs used to generate RKO addback cell lines. All mutant constructs were generated using PCR amplification followed by Gibson assembly and cloned into pLenti CMV Puro DEST[63] using gateway methods. Doxycycline-inducible 3xFLAG–GRP94 (tag inserted after the ER signal sequence) constructs for transient expression in HEK293 cells were cloned by PCR amplification followed by Gibson assembly into pcDNA5/FRT/TO. These were used as templates to generate mutant constructs by QuikChange Site-Directed Mutagenesis (Agilent, 210519, 200514).

**STT3A constructs.** pENTR2B-STT3A-FLAG[4] was used as a template to generate STT3A mutants using PCR amplification followed by Gibson assembly and cloned into pLenti CMV Puro DEST[63] using Gateway methods. All constructs were fully sequenced to confirm accuracy.

## Cell culture

Flp-In T-REx 293 cells (Invitrogen) were maintained at 37 °C and 5% $CO_2$ in DMEM (Corning, MT10013CV) and supplemented with 10% foetal bovine serum (GeminiBio, 900-108) and 100 U ml$^{-1}$ penicillin plus 100 μg ml$^{-1}$ streptomycin (GeminiBio, 400-109). Cells were checked approximately every 6 months for mycoplasma contamination using the Universal Mycoplasma Detection Kit (ATCC, 30-1012K) and were found to be negative.

## HEK293 cell lines

FKBP11 (ref. 26) and STT3A[60] knockout Flp-In T-REx 293 cell lines were generated by CRISPR–Cas9 as previously described. A CCDC134 knockout cell line was generated similarly. In brief, an sgRNA (5′-AG AAGATGTTTGAGGTGAAG-3′) targeting exon 3 of human CCDC134 was designed using the Synthego CRISPR design tool (https://design. synthego.com) and cloned into pSpCas9(BB)-2A-Puro plasmid (PX459; Addgene #62988). Twenty-four hours after transfection, cells were selected under 1 μg ml$^{-1}$ of puromycin (InvivoGen, ant-pr-1) for 72 h. Single cells were isolated by sorting in 96-well plates and allowed to grow clonally. Clonal cells were screened for gene knockout by immunoblotting for CCDC134. Further validation was performed by PCR amplification of ± 200 base pairs of PAM sequence and tracking INDEL by decomposition of Sanger sequencing data using the TIDE web tool[64] and Synthego ICE analysis tool. An STT3B knockout cell line was generated similarly with sgRNA (5′-ACACATCATCTTGCATCTCA-3′) targeting exon 2 of human STT3B and validated by immunoblotting and tracking INDEL.

A stable, doxycycline-inducible 3xFLAG-FKBP11 Flp-In T-Rex 293 cell line was generated in FKBP11 knockout cells as previously described[26]. A stable, doxycycline-inducible 3xFLAG-CCDC134 Flp-In T-REx 293 cell line was generated similarly. Briefly, pOG44 (encoding Flp-recombinase) and pcDNA5 SS$_{ER}$-3xFLAG-CCDC134 at a ratio of 9:1 with 3 μl of TransIT-293 transfection reagent (Mirus Bio, MIR2700) were co-transfected into CCDC134 knockout Flp-In T-REx 293 cells. After 24 h, cells were selected with 100 μg ml$^{-1}$ of hygromycin for 12 days to obtain the stably integrated cell lines. For doxycycline-inducible expression of 3xFLAG–CCDC134, cells were tested in a range of doxycycline concentrations; overexpression was observed without further doxycycline in the culture media.

## RKO cell lines

Stable addback of tagged GRP94, STT3A or CCDC134 were introduced into clonally derived knockout RKO cells using the lentiviral expression system as described previously[4]. Briefly, virus was generated by transfecting HEK293T cells in six-well plates with 200 ng pMD2.G (Addgene), 400 ng psPAX2 (Addgene) and 800 ng of the desired pLenti CMV Puro DEST or pLenti-TRE-rtta3G-BLAST construct using 7 μl of 1 mg ml$^{-1}$ polyethylenimine (Polysciences) per well. Lentivirus-conditioned media was collected after 48 h, filtered through a 0.45-μm filter and 0.5 ml of filtered lentivirus was mixed with 1.5 ml of complete media containing 8 μg ml$^{-1}$ polybrene (MilliporeSigma). The diluted virus was then added to the indicated cells seeded on six-well plates. 24 h post-infection, cells were split and selected with puromycin (2 μg ml$^{-1}$) or blasticidin (10 μg ml$^{-1}$) for 3–7 days or until all of the cells on the control plate are dead. For doxycycline-inducible expression of 3xHA–CCDC134, cells were grown for 24 h in a range of doxycycline concentrations with 5 nM inducing low, near-endogenous expression levels.

## Preparation of rough microsomes

Typically, ten 15-cm dishes of the desired HEK293 cell line were grown to approximately 80% confluency, washed once with ice-cold PBS and collected by scraping in ice-cold PBS. Cells were collected by centrifugation at 1,000 × g for 5 min at 4 °C, flash-frozen in liquid nitrogen and stored at −80 °C for future use. For microsome preparation, the frozen

cell pellet was thawed, resuspended in hypotonic homogenization buffer (10 mM HEPES pH 7.4, 10 mM KOAc, 1 mM MgCl$_2$) and incubated on ice for 30 min. Cells were then mechanically lysed with 50 strokes in a glass Dounce homogenizer chilled in ice and sucrose was added to a final concentration of 200 mM. Whole cells and nuclear fragments were removed by two rounds of centrifugation at 2,000 × g for 10 min at 4 °C and the ER-enriched membrane fraction was collected from the supernatant by centrifugation at 10,000 × g for 15 min at 4 °C. Pelleted membrane fractions were resuspended to a total volume of 1 ml in microsome buffer (50 M HEPES pH 7.4, 250 mM KOAc, 10 mM MgCl$_2$, 250 mM sucrose) supplemented with 0.5× cOmplete EDTA-free protease inhibitor cocktail (Roche, 11836170001) and 1 mM CaCl$_2$. The microsome suspension was treated with 10,000 U ml$^{-1}$ micrococcal nuclease (MNase) (NEB, M0247S) at 37 °C for 25 min followed by 5 U ml$^{-1}$ RQ1 RNase-Free DNase (Promega, M610A) for 5 min at room temperature. Nuclease digestion was quenched with 2 mM EGTA and microsomes were collected by centrifugation at 10,000 × g for 15 min at 4 °C. Pelleted microsomes were resuspended in microsome buffer supplemented with 50 U ml$^{-1}$ SUPERaseIn (Invitrogen, AM2696) to the desired concentration and flash-frozen in liquid nitrogen and stored at −80 °C.

## Sample preparation for selective ribosome profiling

Microsomes were prepared as described above, with the following modifications. All buffers were supplemented with 100 ng ml$^{-1}$ cycloheximide. Samples were kept on ice or at 4 °C throughout, unless otherwise noted. Homogenization buffer was supplemented with 0.5× cOmplete EDTA-free protease inhibitor cocktail (Roche). Nuclear fragments were removed by pelleting twice at 2,800 rpm (1,578 × g) for 10 min. Before MNase treatment, microsomes were resuspended with 1.5 ml of 100 mM KOAc microsome buffer.

Micrococcal nuclease-treated 3xFLAG−CCDC134 and 3xFLAG−FKBP11 microsome pellets (from nine 15-cm dishes) were resuspended with 1 ml microsome buffer supplemented with 2.5% digitonin and 1× cOmplete EDTA-free protease inhibitor cocktail (Roche). The sample was rotated to solubilize for 30 min and insoluble material was removed by pelleting at 13,500 × g for 15 min. The soluble fraction was layered over a 1-ml sucrose cushion (50 mM HEPES pH 7.4, 150 mM KOAc, 1 M sucrose, 5 mM MgCl$_2$, 0.25% digitonin) and centrifuged at 250,000 × g for 2 h in a TLA100.3 rotor. The resulting ribosomal pellet was resuspended in 1 ml microsome buffer supplemented with 1.25% digitonin. A portion of this was reserved for sequencing (which served as the 'input' sample for ribosome profiling) and the remainder was immunoprecipitated in batch using 60 µl anti-FLAG M2 affinity gel (MilliporeSigma, A2220) and end-over-end mixing overnight at 4 °C. The sample was subsequently washed four times with 12 column volumes of microsome buffer supplemented with 0.4% digitonin. The sample was eluted twice with 1.5 column volumes of microsome buffer (with 200 mM KOAc, 0.4% digitonin, 0.5 mg ml$^{-1}$ FLAG peptide). The eluate (which served as the 'IP' sample for ribosome profiling) was collected using a pre-equilibrated spin filter column, frozen in liquid nitrogen and stored at −80 °C ('IP' sample for sequencing).

## Selective ribosome profiling library preparation using Rfoot-seq

To prepare the ribosome profiling library, the RNA concentration in both input and IP samples was determined using the Qubit RNA high-sensitivity assay. To generate RNase footprints, samples with 800 ng of RNA were digested with RNase 1 (LGC Biosearch Technologies, N6901K) at 0.5 U µl$^{-1}$ in a 90-µl reaction at room temperature for 1.5 h. The reactions were stopped by adding 400 µl of TRIzol (Ambion, 15596026), vortexing thoroughly and then adding 100 µl of chloroform. RNA in the aqueous layer was separated by centrifugation at 12,000 × g for 15 min at 4 °C and precipitated overnight by adding 0.1 volumes of 3 M sodium acetate (Invitrogen, AM9740), 10 mg of GlycoBlue (Invitrogen, AM9515) and 1.2 volumes of isopropanol (Fisher Scientific,

BP2618). Purified RNase footprints were subjected to Rfoot-seq library preparation following a previously published method[27,28].

## Rfoot-seq data processing and analyses

Ribosome profiling data were processed by trimming adaptors with Cutadapt v4.1, removing rRNA reads with Bowtie v2.2.6 (ref. 65) and aligning remaining reads to the human hg38 genome and RefSeq-defined transcriptome with TopHat v2.1.0 (ref. 66). Uniquely mapped reads were used for downstream analyses. High-quality reads showing 3-nt periodicity were adjusted to the ribosomal A-site using RibORF (refs. 67–69).

Transcript abundance was quantified as transcripts per million using HTSeq v2.0.3 (ref. 70). Transcript enrichment was calculated as the log$_2$ ratio of IP to input samples, median-centred to all genes and tested for significance using chi-squared tests with Benjamini−Hochberg correction. Positional interaction profiles were generated from codon-level enrichment, computed as the log$_2$ ratio of IP to input at each codon position.

## Sample preparation for cryo-EM

3xFLAG-FKBP11 HEK293 cells were seeded in ten 15-cm dishes and induced with 0.1 ng ml$^{-1}$ doxycycline for 24 h. Microsomes (A260 of 35) were prepared as described above and solubilized in microsome buffer supplemented with 1.5% digitonin and 1× cOmplete EDTA-free protease inhibitor cocktail (Roche). After 1 h at 4 °C on a rotating wheel, the solubilized microsomes were diluted 2× with microsome buffer (150 mM KOAc) and cleared by centrifugation at 15,000 × g for 15 min at 4 °C. The cleared supernatant was incubated overnight at 4 °C with approximately 70 µl of anti-FLAG M2 affinity gel (MilliporeSigma, A2220) pre-equilibrated with wash buffer (50 mM HEPES pH 7.4, 200 mM KOAc, 2 mM MgCl$_2$, 1 mM glutathione, 0.15% digitonin). Resin was washed twice with five column volumes of wash buffer and bound material was eluted with 200 µl of the same buffer supplemented with 0.5 mg ml$^{-1}$ FLAG peptide (APExBIO, A6001) for 10 min at room temperature. The eluate was passed through a pre-equilibrated approximately 30 µm Pierce spin column (Thermo Fisher, 69725), layered over a 300-µl sucrose cushion (50 mM HEPES pH 7.4, 150 mM KCl, 5 mM MgCl$_2$, 1 mM glutathione, 500 mM sucrose, 0.15% digitonin) and centrifuged at 355,000 × g for 1 h at 4 °C in a TLA120.1 rotor (Beckman Coulter; Optima MAX-XP). The ribosome pellet was resuspended in 20 µl of wash buffer supplemented with 0.5 mg ml$^{-1}$ FLAG peptide to a final A260 of 0.57 and immediately used to freeze grids for single-particle cryo-EM.

## Cryo-EM grid preparation and electron microscopy

Quantifoil R1.2/1.3 400 mesh gold grids with 2 nm ultrathin carbon were glow-discharged for 25 s immediately before use. Using a Thermo Fisher Vitrobot Mark IV, 3 µl of freshly prepared sample was applied to the grid, incubated at 22 °C and 100% humidity for 60 s, blotted for 7 s, drained for 0.5 s and frozen in liquid-nitrogen-cooled ethane. Three datasets (811, 1,684 and 4,074 videos) were collected on an FEI Titan Krios at 300 KV with EPU software, using a defocus range from −1.9 to −0.9 µm. Super-resolution videos (pixel size = 0.84 Å) were recorded at a nominal magnification of 53,000× using a K3 BioQuantum direct electron detector (Gatan) and a total electron exposure of 60 e$^-$ Å$^{-2}$ over 52 frames.

## Image processing

All data processing was performed using RELION 5.0 (ref. 71). Videos were motion-corrected using MotionCor2 (ref. 72) with 7 × 5 patches and dose weighting. The contrast transfer function (CTF) was estimated with CTFFIND4.1 (ref. 73). An initial particle set was generated by automated picking using the Laplacian-of-Gaussian method with a diameter range of 200 to 300 Å and a threshold of 1, applied to 4,074 videos binned by 2 during motion correction. Particles were extracted

using a 128-pixel box (5.04 Å per pixel) and subjected to 2D classification ($k = 100$, $T = 2$) to select ribosome-like classes. Selected particles were re-extracted with a 384-pixel box (1.68 Å per pixel) and used for 3D refinement. The resulting map was used to train a Topaz[74] model for autopicking across the full combined dataset (6,569 unbinned videos), yielding 741,983 particles. Topaz-picked particles were extracted with a 256-pixel box (2.52 Å per pixel) and subjected to 2D classification ($k = 100$, $T = 2$) using a 550 Å spherical mask and 100 classes. From these, 26 ribosome-like classes were selected (676,520 particles). 3D refinement was performed using a low-pass-filtered (60 Å) mammalian ribosome reference, followed by 3D classification ($k = 9$, $T = 4$). This yielded four classes corresponding to the secretory translocon (73.8%), one class with weak 40S subunit density (4.5%) and four classes containing poorly aligned particles (21.7%). The 498,812 particles from secretory translocon-containing classes were re-extracted with a 768-pixel box (0.84 Å per pixel) and refined in 3D. Two rounds of 3D refinement and CTF refinement were performed before each dataset was individually polished. The polished particles were then combined for a third round of 3D refinement. To further resolve compositional heterogeneity, focused classification with signal subtraction ($k = 9$, $T = 4$, 416-pixel box, 0.84 Å per pixel) was performed using a mask around the translocon region. This yielded three classes with density for the secretory translocon, GRP94, CCDC134 and FKBP11. These were combined (135,132 particles) and refined with local angular searches. An extra round of focused classification with signal subtraction ($k = 4$, $T = 4$) targeting GRP94, CCDC134, FKBP11 and STT3A yielded two classes: one with (41.2%) and one without (58.8%) density for the three accessory factors. The particle subset containing GRP94, CCDC134 and FKBP11 (55,750 particles) was subjected to local refinement, yielding Map 1 (translocon-only map). These particles were then re-extracted using a 480-pixel box (1.4 Å per pixel) and further refined. A subsequent 3D refinement focused on the 60S ribosomal subunit yielded Map 2 (RTC map). Maps used for model building and display were uniformly low-pass-filtered to different resolutions, low-pass-filtered by local resolution (in RELION) or post-processed using DeepEMhancer[75].

## Model building, refinement and validation

The human 60S subunit from PDB ID 6ZMI and the A/P- and P/E-site tRNAs from PDB ID 6W6L were used as starting models for the ribosome. The TRAP complex from PDB ID 8RJC[30] (mutated to human) and the OST-A complex from PDB ID 8B6L[29] were used as starting models for the secretory translocon.

Initial models for the remaining translocon components (FKBP11, CCDC134 and KCP2) and the GRP94 nascent chain were generated using the ColabFold2 (ref. 76) implementation of AlphaFold2 multimer v3 (ref. 77), as described below (Extended Data Fig. 4). Protein sequences were obtained from UniProt[78].

The quality of each AlphaFold2 multimer prediction was initially assessed by the predicted local-distance difference test (pLDDT), which provides a per-residue confidence score for each subunit, the predicted aligned error (PAE), which provides a confidence measure of the predicted protein–protein interface(s), and by the overall (pTM) and interface (ipTM) predicted TM scores. The AlphaFold2 complexes were also validated by their fit to the experimental density, comparison with previously determined structures and mutational analysis. Low-confidence regions not supported by the cryo-EM density were removed from the model.

All maps were used for model building. The 60S subunit, A/P and P/E tRNAs were placed as rigid bodies into Map 2 and a 43-residue poly-Ala segment of the nascent chain was modelled into exit tunnel density extending into the open SEC61 channel. The model was fit using tightly restrained real-space refinement in Coot[79], including planar and *trans* peptide restraints, Ramachandran restraints and Geman-McClure local distance restraints.

The OST-A complex from 8B6L was placed into Map 1 as a rigid body and adjusted as a single unit using tightly restrained real-space refinement in Coot, as above. The AlphaFold2 model of KCP2 (bound to the N-terminal region of DC2) was placed into the map and adjusted as a single unit (to maintain predicted inter-chain contacts) using tightly restrained real-space refinement in Coot. Weak density in the STT3A catalytic site was visible for a co-purifying LLO ligand, including the pyrophosphate group and portions of the isoprenyl tail and glycan moiety, and was built using PDB ID 8AGC as a guide. Several ordered N-glycans were also built for STT3A and RPN1. Further weak density was visible for what are probably digitonin and lipid molecules, but these were not modelled.

The AlphaFold2 model for SEC61–RAMP4 complex was placed as a rigid body into Map 1 and the complex adjusted as a single unit using tightly restrained real-space refinement in Coot. The displaced plug region of the fully opened SEC61 channel was rebuilt using PDB ID 8RJB as a guide.

Density for the TRAP complex, which was strongest in the lumenal regions, showed substantial displacement from its canonical position adjacent to SEC61. To model this, we divided the model into three separate regions: (1) the TRAPβ,γ,δ membrane and cytosolic segments; (2) TRAPα,β,δ lumenal domains; and (3) the TRAPα TMD and cytosolic tail. The membrane and lumenal regions were fit using tightly restrained real-space refinement in Coot. The TRAPα cytosolic tail was adjusted manually and fit using real-space refinement; no density was observed for the TRAPα TMD, which was not modelled.

The FKBP11–CCDC134 AlphaFold2 model was docked into Map 1 as a rigid body. The C-terminal, positively charged helix of FKBP11 was modelled into weak density adjacent to the ribosome. The flexible C-terminus of CCDC134 was placed into helical density at the OST48–RPN2 interface, guided by an additional AlphaFold2 prediction. The resulting model was adjusted using tightly restrained real-space refinement in Coot.

The AlphaFold2 model of GRP94 (residues 22–97) bound to DC2 and STT3A was placed into Map 1 by superimposing onto the previously placed STT3A subunit. GRP94 residues 73–97 showed poor fit to the helical cryo-EM density and were removed and the three-subunit complex (including GRP94 residues 22–72) was adjusted using tightly restrained real-space refinement in Coot.

Next, the AlphaFold2 model of GRP94 (residues 22–97) bound to CCDC134 was docked into Map 1 by superimposing onto the previously placed model of CCDC134–FKBP11. GRP94 residues 63–92 are predicted by AlphaFold2 to form a single extended helix ('bridge helix'), in excellent agreement with the experimental cryo-EM density. GRP94 residues 22–72 were removed and the two-subunit complex was adjusted using tightly restrained real-space refinement in Coot.

Finally, the globular N-domain of GRP94 was docked into low-resolution density adjacent to CCDC134. Placement was guided by the central helical element of the N-domain and its β-sheet. No density was visible in the GRP94 nucleotide binding site or for the dynamic active-site 'lid', which was removed from the model. Similarly, no density was visible for the GRP94 'charged linker' motif or for the M- and C-domains. The final GRP94 model (residues 22–283) was adjusted as a single complex with CCDC134, FKBP11, DC2 and STT3A using tightly restrained real-space refinement in Coot.

The translocon model was subjected to real-space refinement in PHENIX[80] alone (versus Map 1) or after combining with the 60S model (versus Map 2). Five rounds of global minimization and group B-factor refinement were carried out with Ramachandran and rotamer restraints (rotamer outliers were fixed using the fit option 'outliers' and the target was set to 'fix_outliers'), reference model restraints (starting model) and hydrogen-bonding, base-pair and stacking parallelity restraints applied to the rRNA. Secondary structure restraints were turned off. Final model statistics for the translocon and the 60S-translocon models are provided in Extended Data Table 1. Structure figures were generated with ChimeraX[81].

## Functional analysis in RKO and HEK293 cell lines

RKO whole-cell lysates were prepared in lysis buffer: 50 mM Tris at pH 8.0, 150 mM NaCl, 2% NP-40, 0.25% deoxycholate, 0.1% SDS, 0.5 mM TCEP, 10% glycerol, 1× SIGMAFAST protease inhibitor cocktail (MilliporeSigma) and 1× PhosSTOP phosphatase inhibitor cocktail (Roche). Samples were placed on a shaker for 30 min at 4 °C, centrifuged for 30 min at 20,000 × $g$ at 4 °C and the supernatant was collected and measured by BCA (Thermo Fisher Scientific). To resolve LRP6 and full-length GRP94 glycoforms, samples were run on Novex Tris-Glycine 4 to 12% gels (Thermo Fisher Scientific). For immunoblot analysis of IGF1R, cells were collected by scraping, as IGF1R is sensitive to trypsinization. The resolved proteins were transferred onto nitrocellulose membrane (Bio-Rad Laboratories) using a wet electroblotting system (Bio-Rad Laboratories) followed by immunoblotting.

For EndoH analysis of wild-type and knockout ($\Delta FKBP11$, $\Delta CCDC134$, $\Delta STT3A$, $\Delta STT3B$) HEK293 cells, microsomes were prepared from two 15-cm dishes as described above. A 20-μl aliquot (A260 of 60) was thawed and centrifuged at 10,000$g$ for 10 min at 4 °C. The samples were treated with 2,500 U EndoH (New England Biolabs, P0702L), according to the manufacturer's protocol. Samples were run on home-made tris-glycine 5.5% gels at 4 °C and constant voltage (150 V) or on 4–20% Bio-Rad TGX precast gels, followed by immunoblotting.

For analysis of GRP94 mutants in HEK293 cells, wild-type Flp-In T-REx 293 cells were seeded onto a six-well plate at a density of 200,000 cells per well one day before transfection. A transfection mixture was prepared containing 1 μg pcDNA5-SS-3xFLAG-GRP94, 150 μl Opti-MEM and 3 μl TransIT293, which was incubated at room temperature for 20 min before adding dropwise to each well. After 24 h, cells were collected using cold 1× PBS and centrifuged at 500$g$ for 5 min at 4 °C. The supernatant was discarded and cells were resuspended in 100 μl lysis buffer (50 mM Tris pH 7.4, 150 mM NaCl, 1% NP-40, 0.5% sodium deoxycholate, 0.1% SDS, 0.5 mM PMSF). Samples were incubated on ice for 15 min (with vortexing every 5 min) and the supernatant was collected after centrifugation at 20,000$g$ for 10 min at 4 °C. Samples were analysed by sodium dodecyl sulfate–polyacrylamide gel electrophoresis (SDS-PAGE) and immunoblotting.

## Reporting summary

Further information on research design is available in the Nature Portfolio Reporting Summary linked to this article.

## Data availability

Sequencing data are available at the NCBI Gene Expression Omnibus (GEO) repository with accession number GSE303507. Protein models and cryo-EM maps are available at the RCSB Protein Data Bank with accession codes 9YGY (translocon-only model) and 9N9J (60S subunit and translocon). The corresponding maps are available at the Electron Microscopy Data Bank with accession codes EMD-72945 (Map 1) and EMD-49171 (Map 2). All other data are available in the main text or the supplementary materials.

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

**Acknowledgements** We thank M. Zhao, J. Fuller and the University of Chicago Advanced Electron Microscopy Core Facility (RRID: SCR_019198) for discussions and support. Computational resources were provided by the University of Chicago Research Computing Center (RCC). This work was supported by: National Institutes of Health grants R35GM145374 (to R.J.K.), R01HL161389 and R35GM138192 (to Z.J.), GM118082 and HD113790 (to R.R.), American Cancer Society–Jean Perkins Foundation Postdoctoral Fellowship in Cancer Research PF-20-121-01-TBE (to M.M.), A.P. Giannini Foundation Postdoctoral Fellowship (to M.M.), T32 GM007183 (to F.Z.), and National Science Foundation Graduate Research Fellowships DGE-1746045 (to M.Y.) and DGE-2146755 (to K.D.).

**Author contributions** Conceptualization: M.Y., M.M., R.J., F.Z., R.R., R.J.K. Investigation: M.Y., M.M., R.J., Y.W., Q.L., F.Z., K.D., Z.J., R.R., R.J.K. Visualization: M.Y., M.M., R.J., Y.W., R.J.K. Funding acquisition: Z.J., R.R., R.J.K. Supervision: Z.J., R.R., R.J.K. Writing – original draft: R.J.K. Writing – review and editing: M.Y., M.M., R.J., Z.J., R.R., R.J.K.

**Competing interests** The authors declare no competing interests.

**Additional information**
**Correspondence and requests for materials** should be addressed to Rajat Rohatgi or Robert J. Keenan.

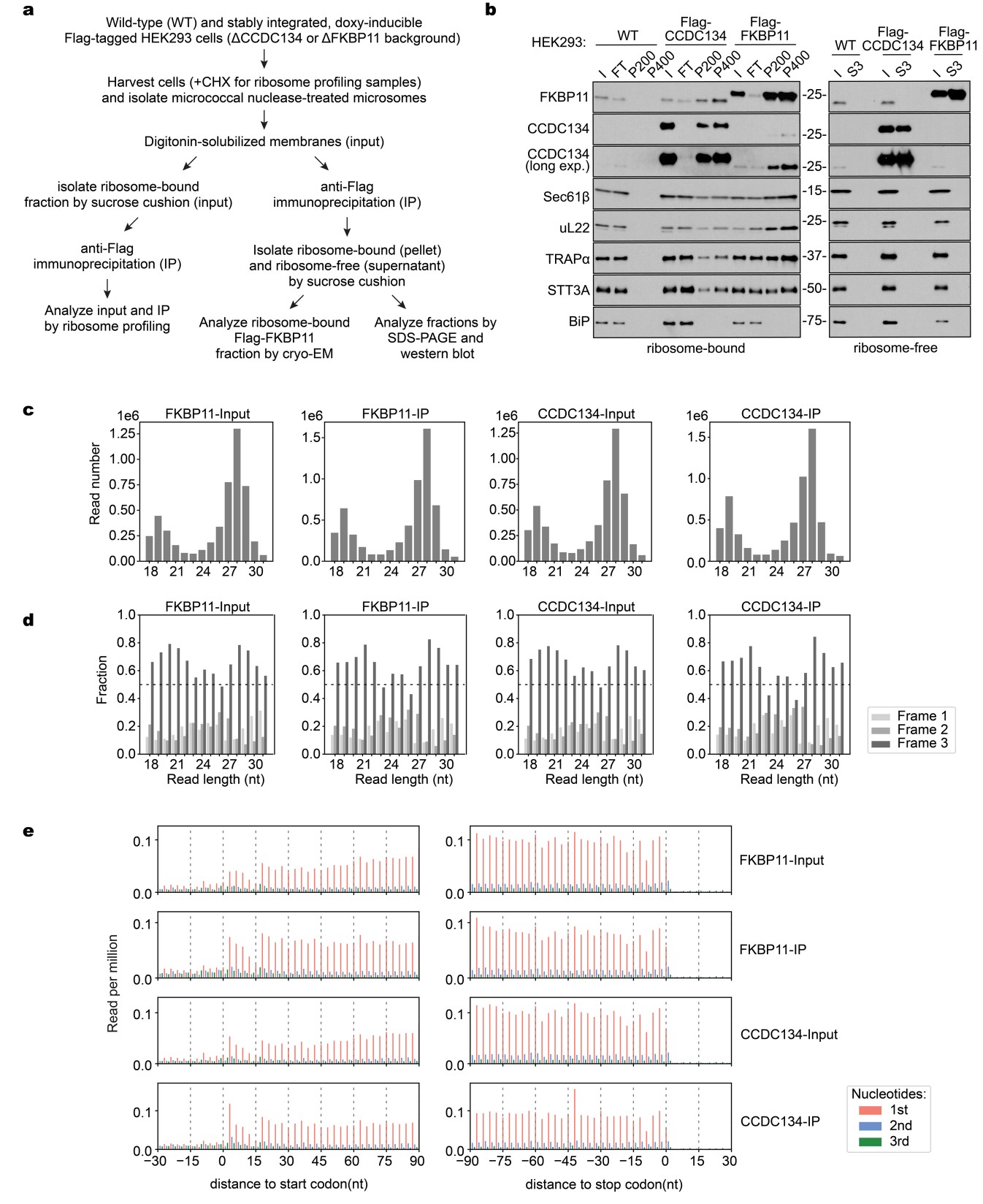

**Extended Data Fig. 1 | Sample preparation strategy and quality control for selective ribosome profiling. a**, Overview of the sample preparation strategy for analysis by western blotting, single-particle cryo-EM and selective ribosome profiling. **b**, Analysis of input (I), flow-through (FT), ribosome-bound (pellet, P, loaded at 200× or 400× input) and ribosome-free (supernatant, S, loaded at 3× input) fractions by SDS-PAGE and immunoblotting. Data are representative of at least two independent experiments. For gel source data, see Supplementary Fig. 1. **c**, Distribution of footprint lengths in coding regions of mRNAs. **d**, Reads were grouped by read lengths (18–35 nt) and the fraction of reads assigned to the three different frames is shown. Strong 3-nt periodicity of read distribution was observed for most footprint sizes. **e**, Ribosomal A-site adjusted read distribution around the start and stop codons of mRNAs.

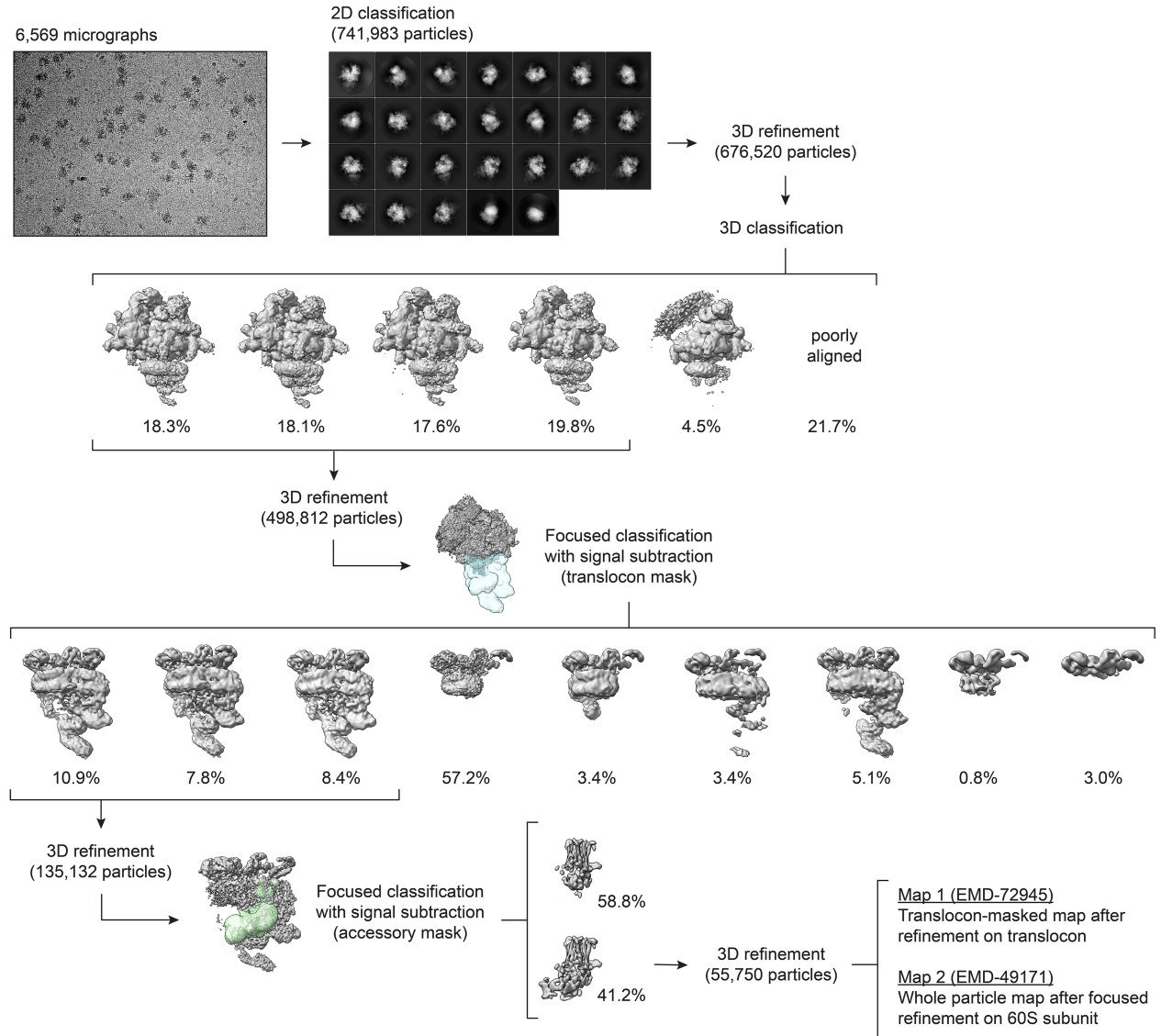

**Extended Data Fig. 2 | Cryo-EM data processing scheme.** Initial rounds of 3D classification yielded four classes containing characteristic density for the secretory translocon (TRAP, SEC61 and OST-A complexes). These were subjected to further 3D refinement and focused classification with signal subtraction using a mask around the translocon, micelle and part of the ribosomal large subunit (cyan). This yielded three main secretory translocon classes, which were refined and subjected to further focused classification with signal subtraction using a mask around GRP94, CCDC134, FKBP11 and STT3A (green). The class with density for GRP94, CCDC134 and FKBP11 was refined to generate final maps of the translocon (Map 1) and the RTC (Map 2).

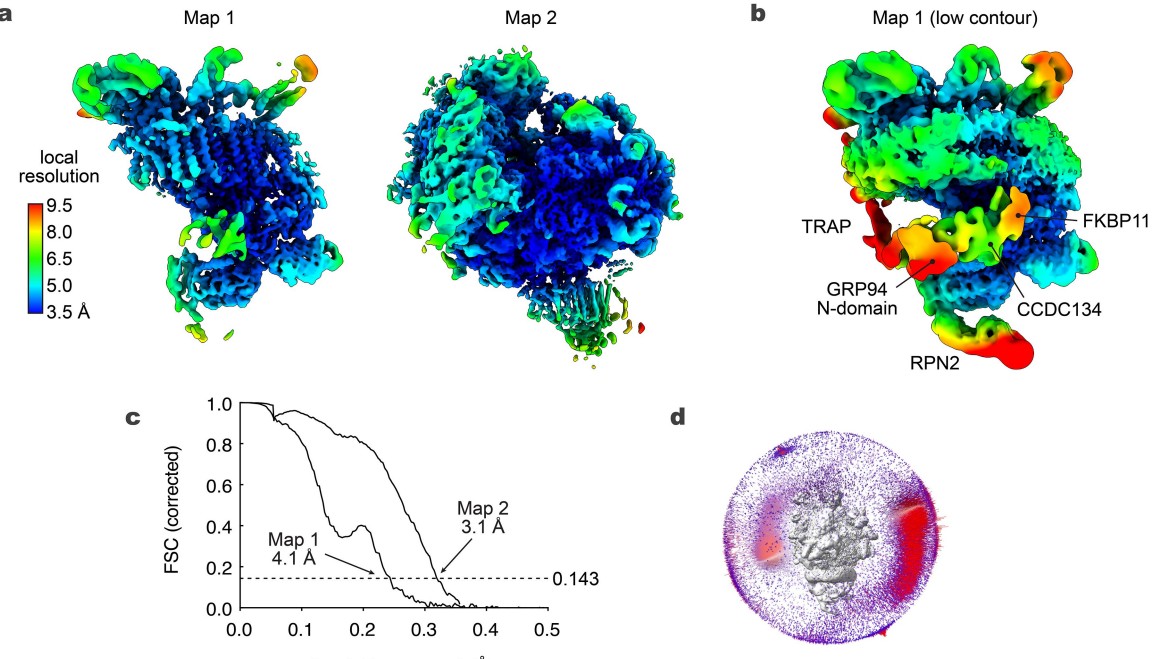

**a** Map 1    Map 2

local
resolution
9.5
8.0
6.5
5.0
3.5 Å

**b** Map 1 (low contour)

TRAP

FKBP11

GRP94
N-domain

CCDC134

RPN2

**c**

**d**

**Extended Data Fig. 3 | Resolution estimates for the GRP94 RTC. a**, Overview of the translocon-only map (Map 1) and whole-particle map after focused refinement on the 60S subunit (Map 2), coloured and low-pass-filtered by local resolution as estimated in RELION 5.0. Maps are coloured on the same scale, from 3.5 to 9.5 Å. **b**, Map 1 shown at a lower contour and coloured by local resolution as in panel **a**. **c**, Fourier shell correlation (FSC) curves (masked). **d**, Angular distribution plot. The height and colour (blue to red) of the bars is proportional to the number of particles in those views.

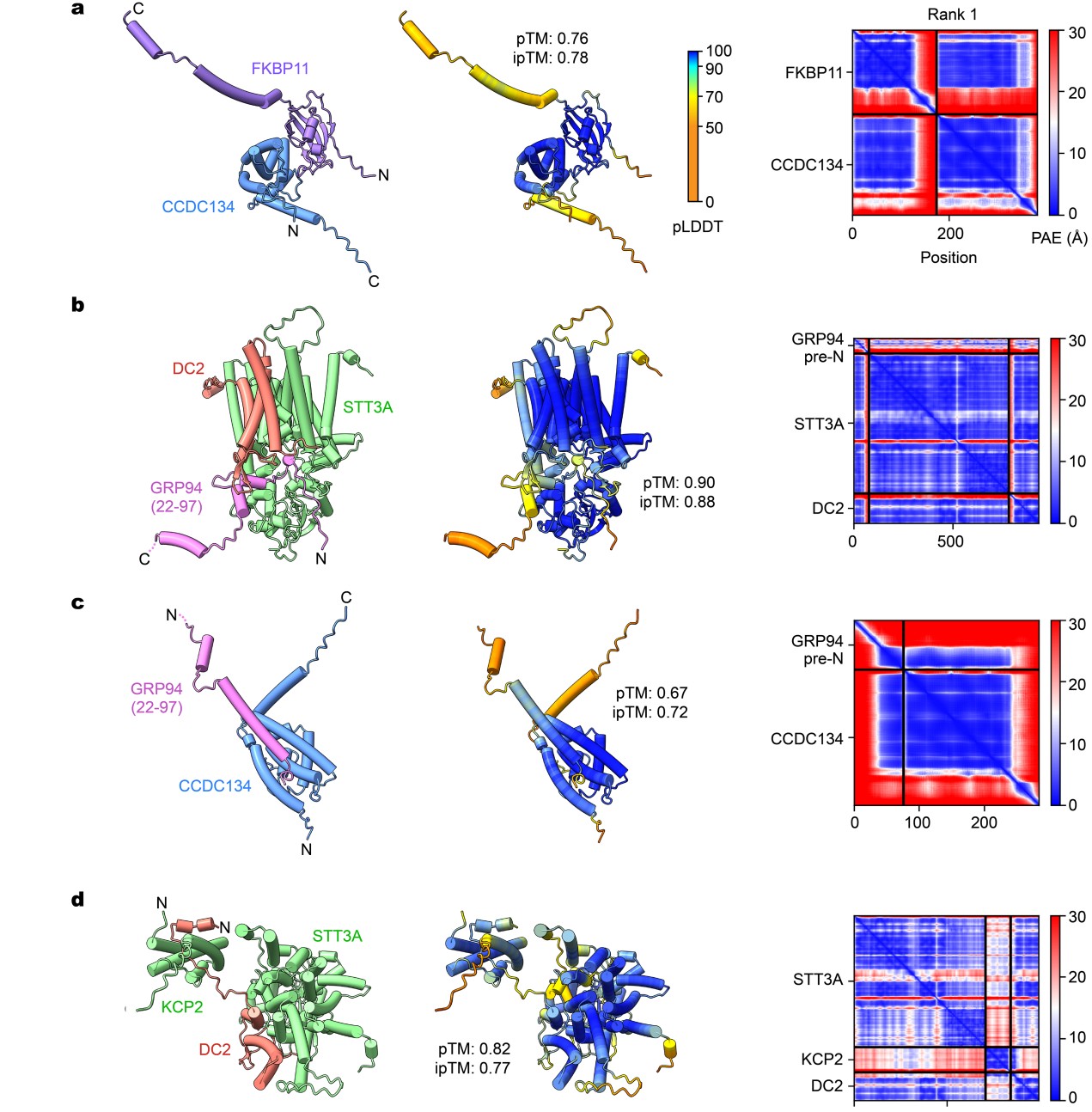

**Extended Data Fig. 4 | AlphaFold predictions.** All predictions were obtained using the ColabFold2 implementation of AlphaFold2 multimer v3 (ref. 77). Models are coloured by chain and by pLDDT. pTM and ipTM scores. Predicted alignment error matrices are also shown. **a**, FKBP11 and CCDC134. **b**, STT3A, DC2 and GRP94 residues 22–97. **c**, CCDC134 and GRP94 residues 22–97. **d**, STT3A, DC2 and KCP2.

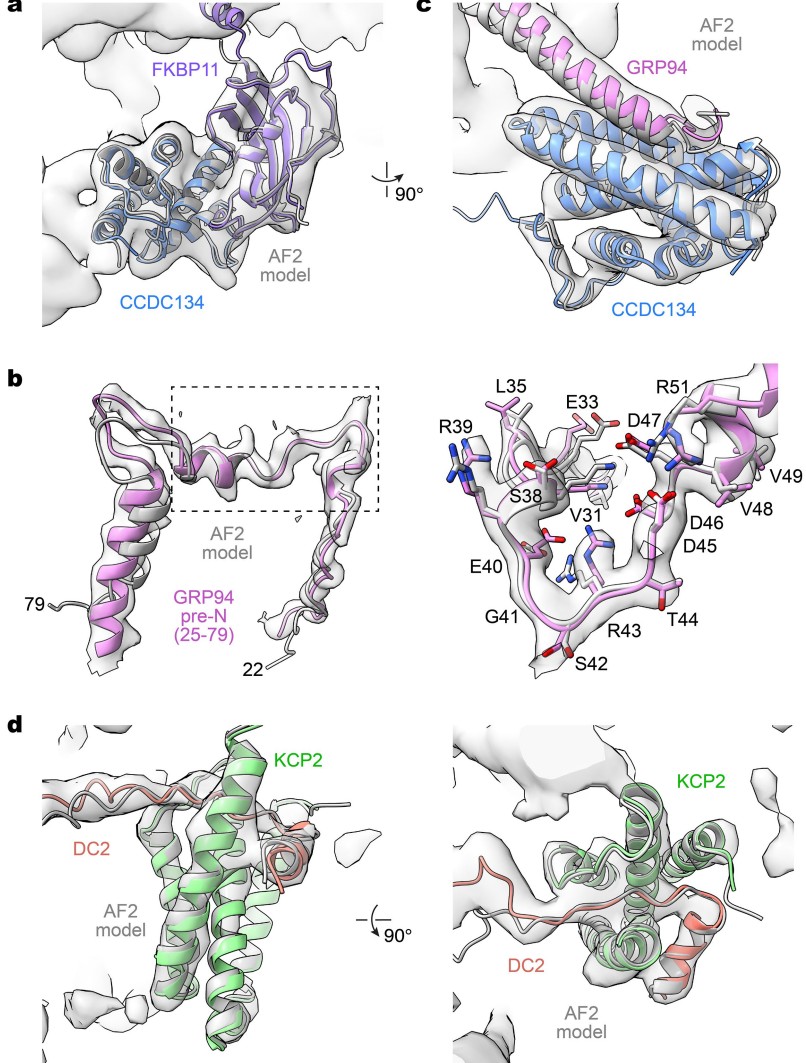

**Extended Data Fig. 5 | Comparison of AlphaFold and final model fits to cryo-EM density. a**, CCDC134 (blue) and FKBP11 (purple) fit to the cryo-EM translocon map low-pass-filtered to 8-Å resolution. In this and subsequent panels, the corresponding AlphaFold2 (AF2) predictions (grey) are superimposed for comparison with the final models. **b**, Overview (left) and close-up (right) views of the OST-A-bound pre-N segment (magenta) fit to the translocon map after low-pass-filtering by local resolution. The AF2 pre-N model (grey) was superimposed by means of the STT3A subunit (not shown for clarity). The correct register of the final model is supported by the clear side-chain density for several residues (for example, R39, S42, R43, T44, R51) and the excellent agreement with the AlphaFold2 model. **c**, Overview of CCDC134 and the GRP94 bridge helix (magenta) fit to the translocon map low-pass-filtered to 6-Å resolution. **d**, Orthogonal views of KCP2 (green) and the DC2 N-terminal extension (salmon) fit to the same map.

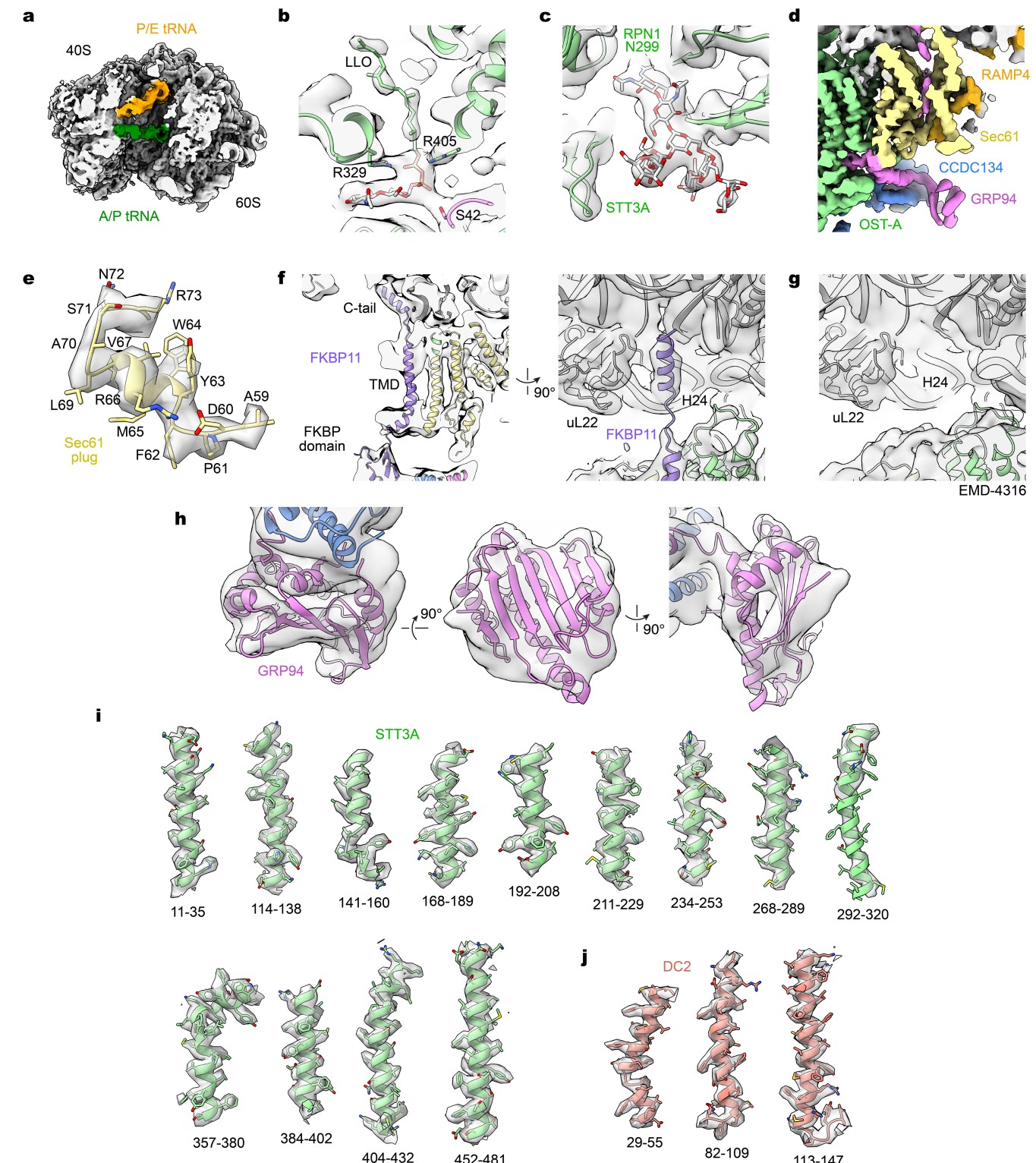

**Extended Data Fig. 6 | Additional views of local map quality and model fits.**
**a**, Slice through the whole-particle map low-pass-filtered by local resolution,
highlighting the hybrid state A/P (green) and P/E (orange) tRNAs on the
ribosome (grey). **b**, Close-up of the translocon map low-pass-filtered by local
resolution, highlighting the LLO ligand bound in the STT3A (green) active site.
A portion of the GRP94 SRT motif (including S42) is also shown (magenta).
**c**, Close-up of an N-linked glycan at RPN1 N299, fit into the translocon map
low-pass-filtered by local resolution. **d**, Slice through the translocon map
low-pass-filtered by local resolution, highlighting nascent GRP94 (magenta)
density in the ribosome (grey) exit tunnel and SEC61 (yellow) channel. The
translocated pre-N and N-domains of nascent GRP94 are also visible in the

OST-A (green) active-site cavity and packed against CCDC134 (blue). **e**, Close-up
of the SEC61 plug helix fit to the translocon map after post-processing using
DeepEMhancer. **f**, View of FKBP11 (purple) fit into the translocon map low-pass-
filtered to 6 Å (left) and a close-up of its C-terminal helix fit to the whole-particle
map low-pass-filtered to 6 Å (right). The ribosome is shown in grey. **g**, Density
for FKBP11 and its C-terminal helix is not visible in other RTC maps, including
EMD-4316, shown here after low-pass-filtering to 6 Å. **h**, Orthogonal views of
the globular GRP94 N-domain (magenta) fit to the translocon map low-pass-
filtered to 6 Å. **i**, Close-up views of the STT3A TMDs fit to the translocon map
after post-processing using DeepEMhancer. **j**, As in panel **i**, but for the DC2 TMDs.

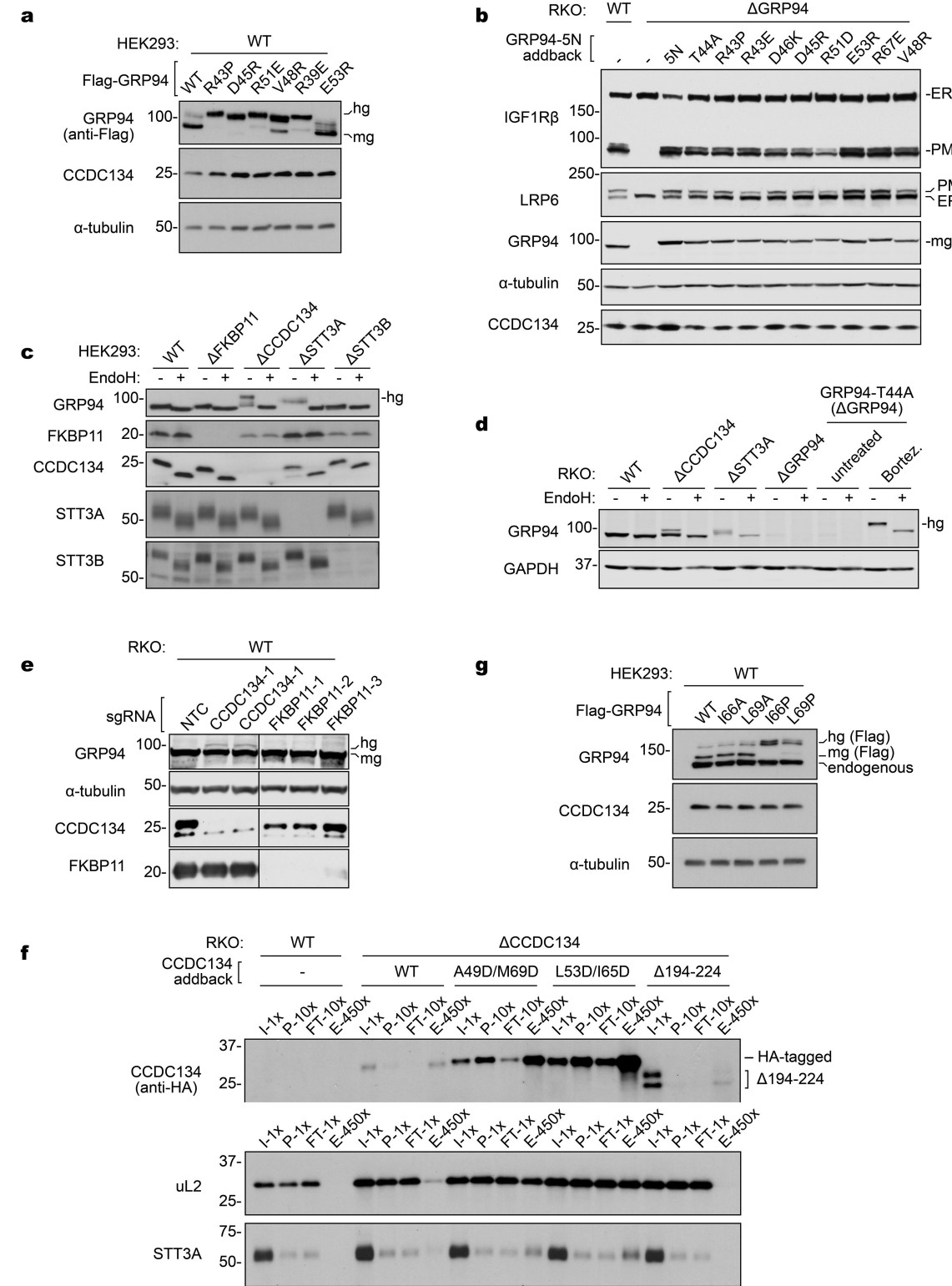

**Extended Data Fig. 7** | See next page for caption.

**Extended Data Fig. 7 | Further characterization in cells. a**, Levels of monoglycosylated (mg) and hyperglycosylated (hg) GRP94 following transient transfection of the indicated variant into HEK293 cells. **b**, ER and plasma membrane (PM) levels of IGFR1β and LRP6 and levels of GRP94-5N (mg) in GRP94 knockout RKO cells stably expressing the indicated GRP94-5N variant. **c**, EndoH analysis of endogenous GRP94 hyperglycosylation (hg) in the indicated HEK293 cell lines. **d**, As in **c** but for the indicated RKO cell lines. The last four lanes show GRP94 knockout RKO cells stably expressing a GRP94 T44A mutant, +/− treatment with the proteasome inhibitor, bortezomib, for 18 h at 1 μM. **e**, Analysis of endogenous GRP94 hyperglycosylation (hg) levels in RKO cell lines expressing sgRNAs targeting the indicated genes (NTC, non-targeting control). **f**, Input (I), ribosome pellet (P), flow-through (FT) and elution (E) fractions from purification of HA-tagged CCDC134 or the indicated mutants stably expressed in CCDC134 knockout RKO cells. The hydrophobic groove mutants (A49D/M69D and L53D/I65D), but not the C-terminal deletion mutant, efficiently assemble with the ribosome-bound translocon. **g**, Levels of monoglycosylated (mg) and hyperglycosylated (hg) GRP94 following transient transfection of the indicated variant into HEK293 cells. The fastest migrating band present in all lanes corresponds to endogenous, monoglycosylated GRP94. Data shown in panels **a**–**g** are representative of at least two independent experiments. For gel source data, see Supplementary Fig. 1.

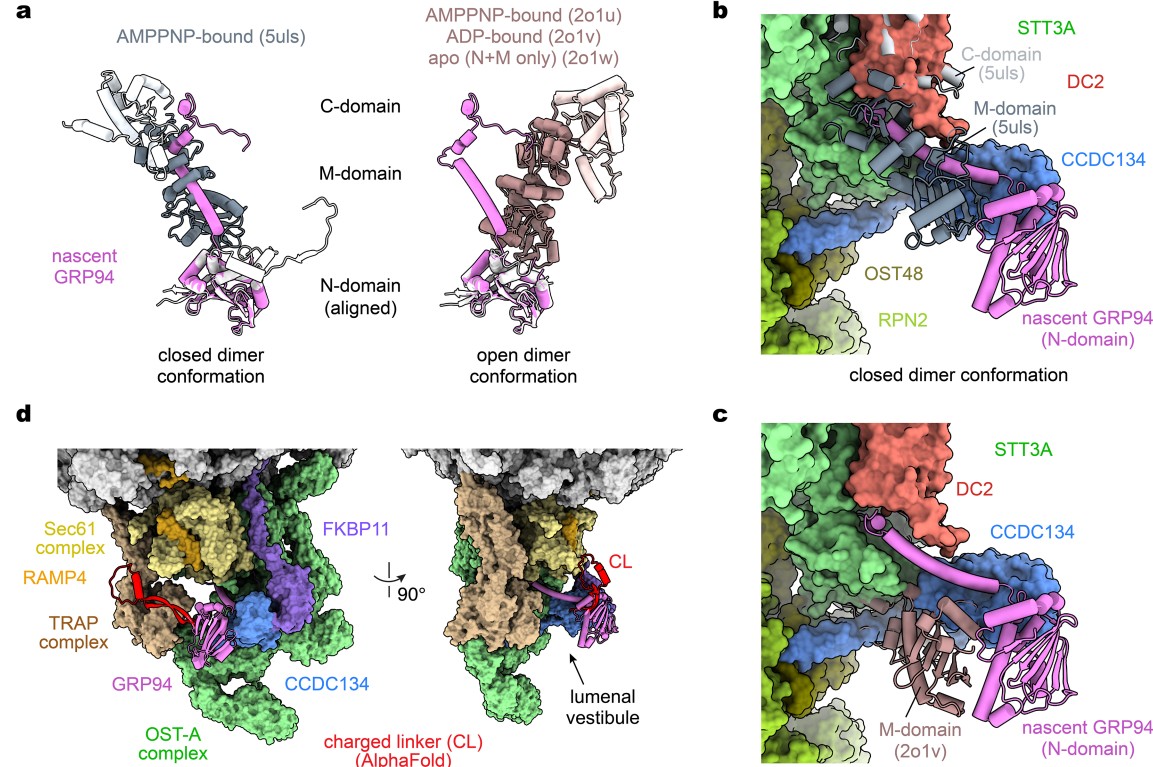

**Extended Data Fig. 8 | The native GRP94 domain interfaces are incompatible with nascent GRP94 binding to OST-A and CCDC134. a**, Structural comparison of folded GRP94 monomers from the indicated experimentally determined structures, aligned on the nascent GRP94 N-domain (magenta) to highlight N–M domain interface flexibility in the closed- and open-dimer conformations. **b**, Overlay of a folded GRP94 monomer (PDB ID 5uls) in a closed-dimer conformation, aligned on the N-domain of nascent GRP94. Severe clashes are observed between the M-domain (dark grey) and STT3A (green), DC2 (salmon), CCDC134 (blue) and portions of the GRP94 pre-N domain (magenta), including the bridge helix. Further clashes are observed between the C-domain and STT3A and DC2. **c**, Overlay of a folded GRP94 monomer (PDB ID 2o1v) in an open-dimer conformation, aligned on the N-domain of nascent GRP94. Severe clashes are observed between M-domain (rosy brown) and STT3A, CCDC134 and RPN1 (not visible in this view). Further clashes are observed between the C-domain and RPN1. FKBP11 was removed for clarity in panels **b** and **c. d**, The charged linker (CL) (red) from an AlphaFold2 model of GRP94 was superimposed onto the structure of nascent GRP94 to highlight its location at the translocon. This antiparallel two-stranded β-sheet projects away from the N-domain, where it may help shield the M- and C-domains as they begin to fold.

**Extended Data Table 1 | Cryo-EM data collection, processing and validation statistics**

| | Translocon-only model<br>PDB 9YGY<br>EMD-72945 (Map 1) | 60S and translocon model<br>PDB 9N9J<br>EMD-49171 (Map 2) |
|---|---|---|
| **Data collection and processing** | | |
| Magnification | 53,000x | |
| Voltage (kV) | 300 | |
| Electron exposure (e$^-$/Å$^2$) | 60 | |
| Defocus range (μm) | -0.9 to -1.9 | |
| Symmetry imposed | C1 | |
| Micrographs | 6,569 | |
| Initial particle images (no.) | 741,983 | |
| Final particle images (no.) | 55,750 | |
| Map resolution at 0.143 FSC (Å) | 4.1 | 3.1 |
| Map resolution range (Å) | 3.9 to 15.5 | 2.8 to 16.7 |
| Pixel size (Å) | 0.84 | 1.40 |
| | | |
| **Refinement and validation** | | |
| Model resolution at 0.5 FSC (Å) | 6.6 | 3.3 |
| Model composition | | |
| Chains | 25 | 88 |
| Non-hydrogen atoms | 37,160 | 183,288 |
| Protein residues | 4,669 | 11,753 |
| Nucleotides | 0 | 4,145 |
| Ligands | 21 | 252 |
| B factors (Å$^2$) | | |
| Protein | 227 | 150 |
| Nucleotide | N/A | 124 |
| Ligand | 162 | 128 |
| RMS deviations | | |
| Bond lengths (Å) | 0.004 | 0.003 |
| Bond angles (°) | 0.593 | 0.588 |
| Validation | | |
| Molprobity score | 1.29 | 1.23 |
| Clashscore | 5.36 | 3.67 |
| Rotamer outliers (%) | 0.00 | 0.02 |
| Ramachandran plot | | |
| Favored (%) | 98.2 | 97.7 |
| Allowed (%) | 1.6 | 2.2 |
| Outliers (%) | 0.1 | 0.1 |

Rajat Rohatgi

# Reporting Summary

## Statistics

For all statistical analyses, confirm that the following items are present in the figure legend, table legend, main text, or Methods section.

| n/a | Confirmed | |
|---|---|---|
| ☐ | ☒ | The exact sample size (*n*) for each experimental group/condition, given as a discrete number and unit of measurement |
| ☐ | ☒ | A statement on whether measurements were taken from distinct samples or whether the same sample was measured repeatedly |
| ☐ | ☒ | The statistical test(s) used AND whether they are one- or two-sided *Only common tests should be described solely by name; describe more complex techniques in the Methods section.* |
| ☒ | ☐ | A description of all covariates tested |
| ☐ | ☒ | A description of any assumptions or corrections, such as tests of normality and adjustment for multiple comparisons |
| ☐ | ☒ | A full description of the statistical parameters including central tendency (e.g. means) or other basic estimates (e.g. regression coefficient) AND variation (e.g. standard deviation) or associated estimates of uncertainty (e.g. confidence intervals) |
| ☐ | ☒ | For null hypothesis testing, the test statistic (e.g. *F*, *t*, *r*) with confidence intervals, effect sizes, degrees of freedom and *P* value noted *Give P values as exact values whenever suitable.* |
| ☒ | ☐ | For Bayesian analysis, information on the choice of priors and Markov chain Monte Carlo settings |
| ☒ | ☐ | For hierarchical and complex designs, identification of the appropriate level for tests and full reporting of outcomes |
| ☐ | ☒ | Estimates of effect sizes (e.g. Cohen's *d*, Pearson's *r*), indicating how they were calculated |

*Our web collection on statistics for biologists contains articles on many of the points above.*

## Software and code

Policy information about availability of computer code

| Data collection | The following software was used for cryo-EM data collection and processing: EPU-3.5.1, RELION-5.0, MotionCor2, CTFFIND4.1, Topaz-0.2.5, DeepEMhancer-0.17 |
|---|---|
| Data analysis | The following software was used for model building, refinement and validation: Colabfold2 implementation of Alphafold2-multimer v3, COOT release 0.9.8.8, Phenix release 1.21.2-5419, and ChimeraX release 1.8. The following software was used to analyze the Rfoot-seq data: Bowtie v2.2.6; TopHat v2.1.0; HTSeq-count v2.0.3; python 3.10; samtools v1.9; deeptools v3.1.1; DeepTMHMM; RibORF 2.0. |

For manuscripts utilizing custom algorithms or software that are central to the research but not yet described in published literature, software must be made available to editors and reviewers. We strongly encourage code deposition in a community repository (e.g. GitHub). See the Nature Portfolio guidelines for submitting code & software for further information.

## Data

Policy information about availability of data

All manuscripts must include a data availability statement. This statement should provide the following information, where applicable:
- Accession codes, unique identifiers, or web links for publicly available datasets
- A description of any restrictions on data availability
- For clinical datasets or third party data, please ensure that the statement adheres to our policy

Sequencing data are available at the NCBI Gene Expression Omnibus (GEO) repository with accession number GSE303507. Protein models and cryo-EM maps are

available at the RCSB Protein Data Bank (PDB ID 9N9J, 9YGY) and the Electron Microscopy Data Bank (EMD-49171, EMD-72945), respectively. All other data are available in the main text or the supplementary materials.

# Research involving human participants, their data, or biological material

Policy information about studies with human participants or human data. See also policy information about sex, gender (identity/presentation), and sexual orientation and race, ethnicity and racism.

| | |
|---|---|
| Reporting on sex and gender | N/A |
| Reporting on race, ethnicity, or other socially relevant groupings | N/A |
| Population characteristics | N/A |
| Recruitment | N/A |
| Ethics oversight | N/A |

Note that full information on the approval of the study protocol must also be provided in the manuscript.

# Field-specific reporting

Please select the one below that is the best fit for your research. If you are not sure, read the appropriate sections before making your selection.

☒ Life sciences   ☐ Behavioural & social sciences   ☐ Ecological, evolutionary & environmental sciences

For a reference copy of the document with all sections, see nature.com/documents/nr-reporting-summary-flat.pdf

# Life sciences study design

All studies must disclose on these points even when the disclosure is negative.

| | |
|---|---|
| Sample size | No sample size calculations were performed. To verify reproducibility, all cell-based experiments were repeated in part or in full, on independent days and with different batches of cell. Sample sizes were sufficient to yield reproducible and consistent results. The cryo-EM structure was determined from a single sample (with 55,750 particles contributing to the final map). |
| Data exclusions | No data were excluded from the analysis. |
| Replication | The Rfoot-seq experiment shown in Fig 1b-f was performed in two biological replicates for Flag-FKBP11 and Flag-CCDC134. Experiments in Fig. 3e,h, Fig. 4e,f, Ext. Data Fig. 1b, and Ext. Data Fig. 7a,b,d,e,f,g were repeated in at least two independent experiments with similar results (except for the GRP94 E53R mutant in Ext. Data Fig. 7a, which was tested once). The experiment in Ext. Data Fig. 7c was repeated twice from whole cell lysate and once from microsomes, with similar results. |
| Randomization | Randomization was not done for functional assays because there is nothing to randomize. |
| Blinding | Blinding is not relevant or practical for the functional and structural work. |

# Reporting for specific materials, systems and methods

We require information from authors about some types of materials, experimental systems and methods used in many studies. Here, indicate whether each material, system or method listed is relevant to your study. If you are not sure if a list item applies to your research, read the appropriate section before selecting a response.

## Materials & experimental systems

| n/a | Involved in the study |
|---|---|
| ☐ | ☒ Antibodies |
| ☐ | ☒ Eukaryotic cell lines |
| ☒ | ☐ Palaeontology and archaeology |
| ☒ | ☐ Animals and other organisms |
| ☒ | ☐ Clinical data |
| ☒ | ☐ Dual use research of concern |
| ☒ | ☐ Plants |

## Methods

| n/a | Involved in the study |
|---|---|
| ☒ | ☐ ChIP-seq |
| ☒ | ☐ Flow cytometry |
| ☒ | ☐ MRI-based neuroimaging |

# Antibodies

| Antibodies used | The following primary antibodies were used: mouse anti-CCDC134 (Santa Cruz Biotechnology, #sc-393390, RRID:AB_3662100, 1:500); mouse anti-GRP94 (R&D Systems, #MAB7606, RRID:AB_3644153, 1:2000); mouse anti-GRP94 (Santa Cruz Biotechnology, #sc-393402, RRID:AB_2892568, 1:2000); rabbit anti-LRP6 (Cell Signaling Technology, #2560, RRID:AB_2139329, 1:1000); mouse anti-α-Tubulin (MilliporeSigma, #T6199, RRID:AB_477583, 1:10000); mouse anti-α-Tubulin (Abcam, #ab11304, RRID:AB_297909, 1:1000); mouse anti-STT3A (Abnova, #H00003703-M02, RRID:AB_530104, 1:1000); rabbit anti-STT3A (Proteintech, #12034-1-AP, RRID:AB_2877818, 1:1000); rabbit anti-IGF1Rβ (Cell Signaling Technology, #9750, RRID:AB_10950969, 1:1000); mouse anti-GAPDH (Proteintech, #60004-1-Ig, RRID:AB_2107436, 1:10,000); rabbit anti-PSAP (GeneTex, #GTX101064, RRID:AB_2037779, 1:1000); mouse anti-HA (GenScript, #A01244, RRID:AB_1289306, 1:1000); rabbit anti-HA (Bethyl, #A191-102, RRID:AB_2891412, 1:2000); rabbit anti-uL22 (Abcepta, #AP9892b, RRID:AB_10613776, 1:1000); rabbit anti-uL2 (Abcam, #ab169538, RRID:AB_2714187, 1:1000); rabbit anti-Sec61β (Cell Signaling Technology, #14648, RRID:AB_2798555, 1:1000); rabbit anti-TRAPα (Millipore Sigma, #HPA011276, RRID:AB_1857503, 1:1000); rabbit anti-STT3B (Proteintech, #15323-1-AP, RRID:AB_2198046, 1:1000); rabbit anti-FKBP11 (Atlas Antibodies, #HPA041709, RRID:AB_10794487, 1:1000); mouse anti-FLAG (Millipore Sigma, #F1804, RRID:AB_262044, 1:1000); mouse anti-BiP (BD Transduction Lab, #610978, RRID:AB_398291, 1:1000). <br><br>The following secondary antibodies conjugated to horseradish peroxidase were used: Peroxidase AffiniPure Donkey Anti-Mouse IgG (H+L) (Jackson ImmunoResearch Laboratories, #715-035-150, RRID: AB_2340770, 1:10,000); Peroxidase AffiniPure Donkey Anti-Rabbit IgG (H+L) (Jackson ImmunoResearch Laboratories, #111-035-144, RRID: AB_2307391, 1:10,000); Peroxidase Donkey Anti-Rabbit IgG (Fc specific) (Sigma Aldrich, #SAB3700863, RRID:AB_3675584, 1:10000); Peroxidase Rabbit Anti-Mouse IgG H&L (Abcam, #ab6728, RRID:AB_955440, 1:10000). Secondary antibodies conjugated to IRDye® 800CW were obtained from LI-COR (IRDye® 800CW Donkey anti-Mouse IgG Secondary Antibody, 1:10,000). |
|---|---|
| Validation | All commercial antibodies were validated by the manufacturers for specificity against human antigen. The following antibodies were additionally validated by knockout or mutant expression experiments in this manuscript: anti-CCDC134, anti-GRP94, anti-LRP6, anti-PSAP, anti-IGF1Rbeta, anti-FKBP11, and anti-STT3A. The following have been further validated in the indicated citations: anti-CCDC134 (PMID: 39509507, 32181939), anti-PSAP (PMID: 39509507, 27383987), anti-STT3A (Proteintech) (PMID: 39509507), anti-STT3A (Abnova) (PMID:36261522), anti-LRP6 (PMID: 39509507), anti-IGF1Rbeta (PMID: 39509507), anti-GRP94 (Santa Cruz Biotechnology) (PMID: 39509507), anti-FKBP11 (PMID: 39259761) |

# Eukaryotic cell lines

Policy information about cell lines and Sex and Gender in Research

| Cell line source(s) | RKO cells were purchased from ATCC, and Flp-In T-REx HEK293 Cell Line are from Invitrogen |
|---|---|
| Authentication | Parental RKO cell line came with a certificate of authentication from the vendor (ATCC) and was used without further validation. Flp-In T-REx 293 cell line was authenticated by the antibiotic resistance markers within its genome. All knockouts were validated by PCR amplification of the genomic locus and immunoblotting for the absence of protein. |
| Mycoplasma contamination | Cells were checked approximately every six months for mycoplasma contamination using the Universal Mycoplasma Detection kit (ATCC), and were found to be negative. |
| Commonly misidentified lines (See ICLAC register) | None used. |

# Plants

| Seed stocks | N/A |
|---|---|
| Novel plant genotypes | N/A |
| Authentication | N/A |

