## [Peer Review File · Nature]

Structural basis of regulated N-glycosylation at the secretory translocon

Corresponding Author: Professor Robert Keenan

Version 0:

Reviewer comments:

Referee #1

(Remarks to the Author)

This manuscript by Yamsek et al. presents a cryoEM-based structural and mechanistic analysis of how the ER-resident chaperone GRP94 regulates its own N-glycosylation during cotranslational translocation. The authors describe how a metazoan-specific N-terminal “pre-N” segment of GRP94 acts as a pseudo-substrate inhibitor of OST-A, and how this inhibitory interaction is stabilized by CDC134 via a hydrophobic groove that binds a non-native α -helix in GRP94. The study also reveals dynamic rearrangement of the TRAP complex and exclusion of OST-B to prevent inappropriate glycosylation of GRP94’s facultative sequons. This work offers insight into a previously unknown layer of ER quality control with potential implications for diseases. While this manuscript presents an intriguing model for regulated N-glycosylation during GRP94 biogenesis at the secretory translocon, a central concern lies in the quality of the cryo-EM map and the structural modeling built upon it. As detailed below, I believe the authors should improve the local resolution and/or adopt a more conservative modeling approach where justified. Additional biochemical validation is also needed to confirm structural assignments and rule out alternative interpretations of glycosylation phenotypes.

Major comments:

1. A major concern in reviewing this manuscript is the quality of the cryo-EM map, particularly regarding key structural features such as the overall modeling of GRP94 and FKBP11 and the local modeling of the nascent peptide within the OST-A active site. Overall, the map densities attributed to GRP94 and FKBP11 lack the resolution and definition needed to support confident modeling of most secondary structure elements. While the densities appear to be proteinaceous rather than noise, they do not resolve side chains or key structural features necessary for accurate residue-level assignment. In the OST-A active site, the density corresponding to the pre-N domain does suggest a continuous segment—potentially a long loop (residues 25–60) followed by a helical element (residues 64–90). However, the absence of clear side chain features makes it difficult to justify residue-specific modeling. Under even the most relaxed modeling standards, this region would more appropriately be represented using an all-alanine poly-peptide rather than the full side-chain-resolved model currently presented. Additionally, the density corresponding to the C-terminal segment near the ribosome is fragmented and low-resolution, making the modeled conformation unreliable.

Because several of the manuscript’s key conclusions rely on precise interactions between the GRP94 pre-N segment and surrounding components of the translocon, the current cryo-EM map resolution and local density quality do not appear to support such detailed structural interpretation. I strongly recommend that the authors improve the local resolution of these key features—particularly the pre-N segment—and revise the atomic model accordingly. In regions where local density is insufficient to support side chain placement, the authors should remove or simplify side chains to avoid overinterpretation and potential misrepresentation of structural details.

2. Protein translation and post-translational modification (PTM) are inherently dynamic processes, and ribosomes represent a heterogeneous population due to their engagement with diverse mRNAs and varying biosynthetic stages. This raises an important mechanistic question: how do the authors determine the cryo-EM structure for a translational and modification complex acting specifically on GRP94 mRNA at a defined stage of synthesis?

The authors note that “GRP94-encoding mRNAs were highly enriched on FKBP11-associated ribosomes.” However, it would strengthen the manuscript to quantify this enrichment—i.e., what proportion of the total mRNAs bound to FKBP11-

associated ribosomes are GRP94 transcripts? Providing such data, for example, via ribosome profiling or quantitative RT-PCR, would help validate the assignment of the observed nascent chain to GRP94.

Additionally, the captured structural state appears to represent a consistent and relatively advanced stage of GRP94 translation. Given the diversity of ribosomal states expected in a native population, it would be helpful if the authors could discuss why this particular segment of GRP94 is so prominently captured. Does GRP94 experience translational pausing at this stage? Are there known features (e.g., rare codons, structural elements in the mRNA, or interactions with CCDC134/FKBP11) that could explain the observed stalling or enrichment of ribosomes at this position?

3. The cryo-EM sample was prepared using FLAG affinity purification from HEK293 cells. While immunoblotting results are shown, it would significantly strengthen the manuscript to include additional biochemical validation of the purified complex. Specifically, SDS-PAGE and/or mass spectrometry analysis would help confirm the composition of the sample used for cryo-EM and verify the presence of the modeled components. In particular, the assignment of the nascent chain within the ribosome and OST-A as GRP94 is central to the study's conclusions. Given the moderate resolution and the absence of clear side chain density in some regions, it is important to provide direct biochemical evidence supporting this assignment.

4. In many of the structural analyses, the authors infer the functional importance of specific protein-protein interfaces—such as the GRP94–CCDC134 interaction—primarily through the effects of interface mutations on GRP94 glycosylation. While this approach is informative, it does not directly confirm the disruption of the physical interaction between GRP94 and CCDC134. Without additional data, it remains possible that the observed hyperglycosylation phenotypes arise from alternative mechanisms, such as global folding defects or impaired interactions with other translocon components. To strengthen the mechanistic conclusions, the authors should provide direct biochemical evidence (e.g., co-immunoprecipitation or pulldown assays) demonstrating that the tested mutations specifically disrupt the GRP94–CCDC134 interaction. Such validation would help distinguish interface-specific effects from broader perturbations in GRP94 biogenesis and ensure that the structural interpretations are supported by physical interaction data.

5. In Extended Data Fig. 3B, the FSC curves for Maps 3 and 4 show drops that suggest the reported 3.4 Å resolution may be overestimated. Since four different maps were used, it would be helpful to specify which map was used for each structural analysis, particularly those focused on the ER membrane and luminal side, at the beginning of the Results section.

6. The authors use differences in migration on Western blots to distinguish mono- and hyperglycosylated forms of GRP94 (e.g., Extended Data Fig. 5A). To more conclusively demonstrate that these band shifts are due to differences in glycosylation, it would be helpful to include a deglycosylation enzyme treatment. This would verify that all observed bands collapse to the same position as the non-glycosylated protein, thereby confirming the glycosylation-dependent mobility shifts.

7. In Extended Data Fig. 5A, both mono- and hyperglycosylated GRP94 are present in the WT lane. This appears to conflict with the model that regulated N-glycosylation protects the nascent chain from hyperglycosylation. Clarification is needed regarding this apparent inconsistency. Additionally, it is unclear where Supplementary Fig. 5A is referenced or located—please verify its inclusion.

Minor points:

1. On the third line of page 3, 'CCCD134' be corrected to 'CCDC134'?

2. The figure legends and labels could benefit from additional clarification to aid reader understanding. For example, in Extended Data Fig. 5B, does the "WT" entry under "GRP94-5N addback" refer to a GRP94 variant lacking all five facultative sequons but without any additional mutations in the pre-N segment? If so, it may be helpful to use a more descriptive term—such as "5N-only"—rather than "WT" to avoid confusion with the true wild-type GRP94.

3. It would be clearer to list the contour level in Figures 1B, 1C, and 1E.

4. On the sixth line of page 4, 'OST-A' be corrected to 'OST-B'?

5. The TRAP complex components (TRAP $\alpha/\beta/\gamma/\delta$) also appear to be affected by limited map quality. In particular, the density in certain regions does not support confident placement of side chains. If local map improvement is not feasible, the authors should consider simplifying the model by omitting side chains—at least in the poorly resolved regions—to prevent overinterpretation and reduce the risk of conveying potentially misleading structural information.

Referee #2

(Remarks to the Author)

I co-reviewed this manuscript with one of the reviewers who provided the listed reports.

Referee #3

(Remarks to the Author)

GRP94 is one of the most abundant molecular chaperones of the endoplasmic reticulum, and in spite of possessing 5 acceptor sequons for co-translational modification by N-linked glycosylation, only a single site is usually modified. In cases that allow the modification of the additional sites can occur, the hyperglycosylated GRP94 is targeted for degradation. A genome-wide CRISPR/Cas9 screen was conducted by some of the authors of the present manuscript, which was recently published in *Science*. It revealed that suppression of hyperglycosylation is dependent on CCDC134, an ER luminal protein, integral membrane components of the OST complex that transfers glycans to specific asparagine residues in client proteins, and the highly conserved N-terminal peptide of GRP94. This earlier study demonstrated interaction of CCDC134 and the OST subunit STT3A with the GRP94 pre-peptide and identified critical residues in the pre-peptide that were essential for interactions with CCDC134 and STT3A and which prevented hyper-glycosylation of GRP94. They proposed this N-terminal peptide acted as a stable pseudo substrate for OST, thereby preventing glycosylation of 4 downstream non-native acceptor sites thereby producing a monoglycosylated chaperone with a long half-life in cells.

This present manuscript employs cryo-EM, is heavily reliant on the AlphaFold program, and includes nascent, translocating

GRP94 (the source of this was not clearly discussed) to provide structural support for the “pseudo substrate” model. The importance of indicated contact sites between the various components were subjected to mutations, and the effects on GRP94 hypermutation were analyzed. A serious hindrance in reviewing this manuscript was the scarceness of experimental details, and in some cases the inclusion of protocols that were not specific to this study.

Figure 1 shows the cryo-EM structure of Sec61, TRAP, and OST-A together with a nascent GRP94 protein associated with these translocon components. The methods state that cells expressing a Flag-tagged version of FKBP11 were used as a source of microsomes, which were subjected to immunoprecipitation for the structural studies. The critical proteins for this study are not clearly resolved, but AlphaFold-derived structures of the three proteins were fit to a luminal density. What is the source of GRP94? I assume the cells are expressing a stalled version of this protein, but there is no mention of this in the methods. The point mutants support the model, and thus in spite of much of the structure being based on AlphaFold, it is very likely that this is the structure that supports suppression of the 4 non-native glycosylation sites.

However, it remains unknown why this very intriguing and complex mechanism exists and under what conditions it would be used. The authors suggest that it could be used to detect misfolded GRP94, but misfolding would have to occur in the first ~40 amino acids of GRP94 to allow the suppressing interactions to occur. Is it possible that amino acid deprivation stress causes pausing and this stabilizes interaction of that “pseudo substrate” peptide? I was also curious as to how the native glycosylation site (N217) is modified, as it is downstream of the blocking peptide. Is this modified by STT3B? And if so, why are N445, N481, and N502 not glycosylated? When does disruption of the pre-peptide:CCDC134:STT3A complex occur? Some time after N502 enters the lumen, I guess? I realize all of these questions are beyond the scope of this study, but maybe some discussion of these points would make the discussion a little more interesting.

Referee #4

(Remarks to the Author)

Jamsek et al. perform a structural analysis of the biogenesis of the ER Hsp90 chaperone GRP94 using cryo-EM and computational modeling. Their study provides deeper mechanistic insight into a recent functional study by Ma et al. (Science, November 2024), which described the glycosylation-inhibitory function of the pre-N domain of GRP94 and its role in hyperglycosylation and premature degradation. Jamsek et al. employed extensive single-particle cryo-EM analysis and AlphaFold2 modeling to study nascent GRP94 bound to a customized ER translocon, and they validated some of their findings using selected mutations.

The authors present a structure of the GRP94-translating ribosome bound to a secretory translocon composed of the known sub-complexes Sec61, OST-A, and TRAP, along with the substrate-specific components CCDC134, KCP2, and FKBP11. The involvement of FKBP11, not previously described, proved key to purifying this complex via pulldown from HEK cells. The structure reveals the roles of FKBP11 and CCDC134 and shows how GRP94 partially folds into a non-native state to form a “cap” over the OST-A catalytic site, thereby inhibiting glycosylation activity. FKBP11 and CCDC134 stabilize this interaction through extensive contacts.

A secondary, minor structural finding is the displacement of the TRAP complex, creating space that promotes correct GRP94 folding and limits excessive glycosylation during co-translational folding by OST-B.

It is exciting to see the structure of the GRP94-folding translocon, which complements the recently uncovered mechanism for substrate retention at OST-A for folding, and contributes to the emerging concept of the translocon as a highly dynamic, substrate-tailored entity. The manuscript is well-written and nicely illustrated. The authors made expert use of AF2 models to interpret relatively low-resolution cryo-EM data. However, the limited resolution makes it challenging for this reviewer to fully assess the quality of the models and the conclusions drawn. Several points should be addressed to make this manuscript suitable for publication:

- The EM map is based on an FKBP11 pulldown. While the occurrence of GRP94 is somewhat unexpected, numerous examples exist where chaperone pulldowns yielded exciting substrate-containing structures. Nevertheless, additional orthogonal evidence confirming that GRP94 is a component of the imaged complex would strengthen the work, given GRP94's central role and the relatively low resolution and conformational differences compared to mature GRP94. For example, intensity-based quantification from mass spectrometry could demonstrate the high abundance of GRP94. Alternatively, RNAseq (as used by the Keenan lab in McGilvray et al., 2020) could help establish GRP94's prevalence. An even more informative experiment—perhaps beyond the scope of this study—would be FKBP11 pulldown in a Δ GRP94 background to assess whether the complex still forms and how this affects structure and substrate binding.
- The single-particle cryo-EM reconstruction of the ribosome-bound GRP94-translocon appears sound but is of limited resolution. While a global resolution of ~3.4 Å is reported, it is dominated by the ribosome and therefore provides limited insight into the translocon region. Local resolution appears substantially lower, especially for GRP94/FKBP11/CCDC134/KCP2/DC2 (~8–11 Å according to Extended Data Fig. 3C). This should be explicitly clarified in the text.
- All model statistics (Extended Data Table 1) include the ribosome, which skews the interpretation of structural quality. Model statistics should be reported separately for the translocon, especially GRP94/FKBP11/CCDC134/KCP2/DC2. Model-to-map cross-validation metrics would also be highly relevant.
- It would be helpful if the authors included a ChimeraX session containing all reported maps and fitted models to allow a more detailed review.
- Due to the low resolution, AF2 modeling is crucial for assignment. While this is acceptable, the assignment is difficult to evaluate because only refined models are shown—which necessarily fit the maps. Pre- and post-refinement models superposed onto the maps should be included as supplementary material to illustrate the modeling process.
- Refinement at very low resolution is prone to overfitting. The authors wisely mention using “tightly restrained” real-space

refinement, but the specific methodology and parameters used should be clearly defined in the Materials and Methods section.

- The fit of the pre-N segment to cryo-EM density (Fig. 2c) is not very convincing regarding the precise register of the model, as sidechains, including bulky residues such as F76 and F78, are unresolved. Given the modest resolution, the detailed panels E–G also appear somewhat speculative.
- TRAP is visible in a previously published multipass-translocon map by McGilvray et al. (eLife, 2020), though it was not modeled. In this map of the multipass-translocon, the TRAP position also deviates substantially from later reconstructions of the same complex from other groups (e.g., Smalinskaitė et al., Nature 2022; Gemmer et al., Nature 2023). The authors should comment on whether specific solubilization protocols may possibly have altered TRAP's position in both cases.
- The assignment of KCP2 is particularly difficult to follow, as the AlphaFold2 multimer prediction primarily involves the very flexible N-terminal region of OSTC, making the tracing possibly challenging.
- Although AF2 modeling guided most of the structure assignments, the basis for FKBP11's ribosome binding is unclear. Given the resolution, assignment based solely on the EM data seems insufficient and should be better explained.

Referee #5

(Remarks to the Author)

I co-reviewed this manuscript with one of the reviewers who provided the listed reports.

Version 1:

Reviewer comments:

Referee #1

(Remarks to the Author)

The revisions have addressed the main concerns from the previous review, particularly with respect to cryo-EM map quality, model interpretation, and biochemical validation. The structural assignments are now presented more conservatively, and I find the revised manuscript suitable for publication.

I have only a few minor suggestions that could improve clarity in the final version:

Fig. 1f – The positional analysis shows distinct interaction profiles of GRP94 with CCDC134 and FKBP11. The GRP94–FKBP11 interaction reaches a high level rapidly before the synthesis of residue ~200, whereas the GRP94–CCDC134 interaction increases more gradually, peaking between residues ~350–400. Please consider clarifying your interpretation of these differing interaction kinetics and discussing the possible mechanistic basis for this temporal offset in recruitment.

Lines 114–116 and 159–160 – Consider referencing the relevant figures that show the densities for the LLO donor and the N-glycans, so that readers can more easily locate the supporting structural evidence.

Referee #2

(Remarks to the Author)

I co-reviewed this manuscript with one of the reviewers who provided the listed reports.

Referee #3

(Remarks to the Author)

The authors have answered my questions in this revised version and I am satisfied by their responses. I support publication of this manuscript in Nature.

Referee #4

(Remarks to the Author)

The additionally provided data greatly strengthen the revised manuscript and support the conclusions. The provided chimeraX network as well as the PDB report also help to better analyze strengths and weaknesses of the data. The cryo-EM data remain unexpected and exciting. However, the resolution of the translocon is mostly limited to the secondary structure level, which makes circumstantial evidence from alphafold and ribosome profiling key to support the conclusions. The EM data still appear overinterpreted in some respects, and consistency of ribosome sequencing and image classification must be clarified.

Specific points:

- The authors nicely show significant enrichment of Grp94-coding mRNA in the pulldowns. Ultimately the RNA profiling should, however, support the assignment of cryo-EM density. Hence, an attempt should be made to directly correlate transcript abundance with cryo-EM image classification. The key question is: Does the RNAseq analysis at least approximately support the assignment of Grp94? According to the EM data (Ext. Fig. 2) > 10% of the total ~500k ribosome particles have GRP94 bound. Does this approximately match the RNA-seq quantification? In other words: is the ribosome profiling supporting the notion that 1/10 of all (presumably all elongating) ribosomes in the sample synthesize Grp94?

- This reviewer appreciates the effort to improve the resolution of the reconstruction, but the visualization of side chains in Fig. 1c/d remains unconvincing. The Q-scores in the PDB report (well below 0.5 for even for the pre-N segment of Grp94) do not support compelling fit of side chains. Exactly this, however, seems to be message of these panels, which should be avoided. What makes the assignment of density to Grp94 and the backbone model nevertheless solid in this reviewer's opinion is the very good agreement of the AF2 backbone model and the EM density (Extended Data Fig. 5b). Hence, this reviewer suggests exchanging these two panels, i.e., making the comparison to AF a main figure as it is essential for the conclusions.

- The modeling of the FKBP11 TM and cytosolic domains from the EM map are weak. The TM domain is not resolved at all (as opposed to the TMs is binds to according to the model). A short helix in the cytosol is resolved, yet tracing from lumen to cytosol is impossible. If the authors want to retain the ribosome-bound parts of FKBP11, the reviewer's suggestion is the following: (i) remove the TM domain from the PDB model, as it is not supported by the EM data. (ii) provide a difference map of a (previously published) FKBP11-free translocon density with their density to provide some EM-based support for this assignment. Again, the EM data alone are not sufficient for assignment, but comparative analysis may make it plausible.

Referee #5

(Remarks to the Author)

I co-reviewed this manuscript with one of the reviewers who provided the listed reports.

Version 2:

Reviewer comments:

Referee #4

(Remarks to the Author)

The revision clarified the remaining issues, yet documentation and data deposition should be updated accordingly.

Specific points:

- The clarification on Q-scores is supporting the conclusion. Yet, the deposited data are confusing. If the discussed atomic model is built into the translocon-focused EM data this should also be the relevant one for PDB and EMDB, and not the ribosome-translocon map. The currently deposited map (EMD-49171) is the full ribosome-translocon map – yet, the atomic model PDB 9N9J has poor Q-scores for the relevant parts of this map. Q-scores are satisfactory for the translocon-focused EM, is not deposited to the EMDB (the manuscript does not indicate so). Hence, this reviewer strongly suggests uploading the translocon-focused map onto the EMDB and to link the translocon PDB to this file (rather than EMD-49171). The full map ('Map 2') may be nice to have for bragging with resolution, but the translocon-focused map ('Map 1') is the one that supports the conclusions.

- The Extended Data Table 1 also requires modification: in the title a single EMDB code is mentioned - yet two maps. Map 1 must also be submitted to the EMDB and the table must be modified with both EMDB codes.

- It should also be clarified in the materials and methods section which of the two maps was used for model building – presumably Map1.

- The provided extended figure 6F is helpful. At this low resolution this reviewer still recommends adding the EM density of a published 'FKBP11-free' ribosome-translocon as a reference to support the assignment of the C-tail, which is minimal work in ChimeraX.

Referee #5

(Remarks to the Author)

I co-reviewed this manuscript with one of the reviewers who provided the listed reports.

We thank the reviewers for their feedback. A summary of the key revisions and detailed responses to each comment are provided below.

- We performed selective ribosome profiling on FKBP₁₁ and CCDC₁₃₄, which addresses several questions raised by the reviewers, including: (1) identifying GRP₉₄ as the most abundant transcript present in the affinity-purified sample used for structural analysis; and (2) demonstrating when FKBP₁₁ and CCDC₁₃₄ arrive and depart from nascent GRP₉₄. [Fig. 1 and Ext. Data Fig. 1]
- We obtained improved cryo-EM maps by re-processing the data to full resolution and performing a more extensive particle classification. [Ext. Data Fig. 2 and 3]
- We added examples of model-map fits, including overlays of all AlphaFold predictions onto the final models to highlight the agreement with each other and the corresponding cryo-EM maps. [Ext. Data Fig. 5 and 6]
- We added an experiment that tests the ability of CCDC₁₃₄ hydrophobic groove and C-terminal deletion mutants to assemble at the translocon. [Ext. Data Fig. 7f]

Reply to Reviewers 1 and 2

A major concern in reviewing this manuscript is the quality of the cryo-EM map, particularly regarding key structural features such as the overall modeling of GRP₉₄ and FKBP₁₁ and the local modeling of the nascent peptide within the OST-A active site. Overall, the map densities attributed to GRP₉₄ and FKBP₁₁ lack the resolution and definition needed to support confident modeling of most secondary structure elements. While the densities appear to be proteinaceous rather than noise, they do not resolve side chains or key structural features necessary for accurate residue-level assignment. In the OST-A active site, the density corresponding to the pre-N domain does suggest a continuous segment—potentially a long loop (residues 25–60) followed by a helical element (residues 64–90). However, the absence of clear side chain features makes it difficult to justify residue-specific modeling. Under even the most relaxed modeling standards, this region would more appropriately be represented using an all-alanine poly-peptide rather than the full side-chain-resolved model currently presented. Additionally, the density corresponding to the C-terminal segment near the ribosome is fragmented and low-resolution, making the modeled conformation unreliable. Because several of the manuscript's key conclusions rely on precise interactions between the GRP₉₄ pre-N segment and surrounding components of the translocon, the current cryo-EM map resolution and local density quality do not appear to support such detailed structural interpretation. I strongly recommend that the authors improve the local resolution of these key features—particularly the pre-N segment—and revise the atomic model accordingly. In regions where local density is insufficient to support side chain placement, the authors should remove or simplify side chains to avoid overinterpretation and potential misrepresentation of structural details.

The concerns raised here (and by Rev 4/5) focus on the resolution of the cryo-EM maps used to generate the model. We addressed this by re-processing the data to full resolution and performing a more extensive particle classification, yielding maps with improved density for most of the translocon components and GRP₉₄. To help the reader better assess the structure quality, we added more examples of model-map fits, including overlays of all AlphaFold predictions onto the final models. The pre-N domain sequence register is supported by sidechain densities for multiple residues (e.g., R₃₉, S₄₂, R₄₃, T₄₄, R₅₁) and by close agreement with the AlphaFold model (see Fig. 3d and Ext. Data 5b). Importantly, while the new maps show greater detail in many key regions, the updated model and our interpretation remain largely unchanged.

Density for the nascent GRP₉₄ polypeptide in the ribosome exit tunnel and Sec61 channel is indeed fragmented and low-resolution. This is because the natively isolated sample (via a Flag-tag on FKBP₁₁) contains a mixture of different length GRP₉₄ translation intermediates. Density for the pre-N and N-domains is visible because they are present in the majority of ribosomes recovered by FKBP₁₁ affinity purification, and their position and structure do not change appreciably during synthesis of downstream regions. In contrast, density in the ribosome exit tunnel and Sec61 channel is weak, reflecting the mixture of different GRP₉₄ lengths (and thus sequence in the tunnel) present in the sample. Accordingly, this region of the nascent chain is modeled as poly-alanine.

Protein translation and post-translational modification (PTM) are inherently dynamic processes, and ribosomes represent a heterogeneous population due to their engagement with diverse mRNAs and varying biosynthetic stages. This raises an important mechanistic question: how do the authors determine the cryo-EM structure for a translational and modification complex acting specifically on GRP94 mRNA at a defined stage of synthesis? The authors note that "GRP94-encoding mRNAs were highly enriched on FKBP11-associated ribosomes." However, it would strengthen the manuscript to quantify this enrichment—i.e., what proportion of the total mRNAs bound to FKBP11-associated ribosomes are GRP94 transcripts? Providing such data, for example, via ribosome profiling or quantitative RT-PCR, would help validate the assignment of the observed nascent chain to GRP94. Additionally, the captured structural state appears to represent a consistent and relatively advanced stage of GRP94 translation. Given the diversity of ribosomal states expected in a native population, it would be helpful if the authors could discuss why this particular segment of GRP94 is so prominently captured. Does GRP94 experience translational pausing at this stage? Are there known features (e.g., rare codons, structural elements in the mRNA, or interactions with CCDC134/FKBP11) that could explain the observed stalling or enrichment of ribosomes at this position?

As requested, we now provide quantitative measures of GRP94 transcript abundance and enrichment in ribosome complexes that co-purify with FKBP11 and CCDC134 (Fig. 1b-e). These data confirm our previous observation (DiGuilio et al., 2024) that GRP94 is the most abundant transcript in samples purified by FKBP11, and we now show this is also true for samples purified by CCDC134.

As noted above, the reason we can visualize the pre-N and N-domains is because they are present in the majority of ribosomes recovered by FKBP11 affinity purification, and their position and structure (bound to OST-A and CCDC134) do not change appreciably during synthesis of the downstream ~520 residues. This is consistent with the selective ribosome profiling, which shows that FKBP11 and CCDC134 engage GRP94 early and remain bound until synthesis is complete (Fig. 1f).

The cryo-EM sample was prepared using FLAG affinity purification from HEK293 cells. While immunoblotting results are shown, it would significantly strengthen the manuscript to include additional biochemical validation of the purified complex. Specifically, SDS-PAGE and/or mass spectrometry analysis would help confirm the composition of the sample used for cryo-EM and verify the presence of the modeled components. In particular, the assignment of the nascent chain within the ribosome and OST-A as GRP94 is central to the study's conclusions. Given the moderate resolution and the absence of clear side chain density in some regions, it is important to provide direct biochemical evidence supporting this assignment.

We apologize for the confusion. We performed the requested experiment in an earlier study (DiGuilio et al., MBoC 2024, Fig. 2): affinity purification of Flag-tagged FKBP11 followed by mass spectrometry analysis of the ribosome-bound fraction. The most abundant proteins identified in that experiment included ribosomal subunits, CCDC134, GRP94 (HSP90B1) and components of the Sec61, OST-A and TRAP complexes. In Ext. Data Fig. 1b of the current manuscript we show SDS-PAGE/western blotting of Flag-tagged FKBP11 and CCDC134 samples, which demonstrates co-purification of the same set of ribosome-translocon components. These data are fully consistent with our new selective ribosome profiling results, AlphaFold modeling of GRP94 interactions with OST-A and CCDC134, the experimental cryo-EM density, and our mutational analysis.

In many of the structural analyses, the authors infer the functional importance of specific protein-protein interfaces—such as the GRP94–CCDC134 interaction—primarily through the effects of interface mutations on GRP94 glycosylation. While this approach is informative, it does not directly confirm the disruption of the physical interaction between GRP94 and CCDC134. Without additional data, it remains possible that the observed hyperglycosylation phenotypes arise from alternative mechanisms, such as global folding defects or impaired interactions with other translocon components. To strengthen the mechanistic conclusions, the authors should provide direct biochemical evidence (e.g., co-immunoprecipitation or pulldown assays) demonstrating that the tested mutations specifically disrupt the GRP94–CCDC134 interaction. Such validation would help distinguish interface-specific effects from broader perturbations in GRP94 biogenesis and ensure that the structural interpretations are supported by physical interaction data.

We tested the ability of CCDC134 hydrophobic groove and C-terminal deletion mutants to assemble at the translocon by affinity purification (Ext. Data Fig. 7f). These experiments were performed in CCDC134 knockout RKO cells stably expressing the HA-tagged mutant proteins, as in Fig. 4e. The two hydrophobic groove mutants (A49D/M69D and L53D/I65D) were recovered in the ribosome-bound fraction, suggesting

that the mutants fold and assemble at the translocon, but are deficient in their interaction with GRP94. This likely destabilizes the inhibitory binding of the pre-N domain to OST-A, leading to the observed GRP94 hyperglycosylation phenotype.

In contrast, the C-terminal deletion mutant ($\Delta 194-224$), which was soluble and expressed well, was not recovered in the ribosome-bound fraction. This suggests that the C-terminus helps stabilize CCDC134 at the translocon during GRP94 synthesis—consistent with the structural observation that the CCDC134 C-terminus packs in a cleft between RPN2 and OST48,

This experiment highlights different ways that CCDC134 function can be perturbed: (1) by interfering with CCDC134 recruitment (C-terminal deletion mutant) to the translocon; or (2) by disrupting the $\alpha N1$ -CCDC134 interface without affecting recruitment. In each case, the result is GRP94 hyperglycosylation. We have added these data to Ext. Data Fig. 7f, and modified the text accordingly.

In Extended Data Fig. 3B, the FSC curves for Maps 3 and 4 show drops that suggest the reported 3.4 Å resolution may be overestimated. Since four different maps were used, it would be helpful to specify which map was used for each structural analysis, particularly those focused on the ER membrane and luminal side, at the beginning of the Results section.

These data were collected in super-resolution mode (pixel size=0.84 Å) at a nominal magnification of 53,000x, and initially processed using the binned pixel size (1.68 Å), corresponding to a Nyquist limit of ~3.36 Å. However, FSC curves for maps 3 and 4 suggested that higher-resolution information was being truncated at this sampling rate. We addressed this by re-processing the data at the smaller pixel size (super-resolution), which, combined with deeper classification, resulted in significantly improved maps. These details are presented in Ext. Data Fig. 2 and 3, and the methods have been updated. We have also revised the figure legends to specify which map was used for each structural analysis.

The authors use differences in migration on Western blots to distinguish mono- and hyperglycosylated forms of GRP94 (e.g., Extended Data Fig. 5A). To more conclusively demonstrate that these band shifts are due to differences in glycosylation, it would be helpful to include a deglycosylation enzyme treatment. This would verify that all observed bands collapse to the same position as the non-glycosylated protein, thereby confirming the glycosylation-dependent mobility shifts.

Glycosidase treatment and N-glycoproteomics was used in our prior publication (Ma et al., 2024) to demonstrate the mono- and hyper-glycosylated forms of GRP94 in different genetic backgrounds and in response to different mutations. Our current manuscript shows EndoH analysis of GRP94 hyperglycosylation in wild-type and several different knockout HEK293 cell lines (now shown in Ext. Data Fig. 7c), and in this revision we have added a new panel showing a similar experiment in different RKO cells (Ext. Data Fig. 7d). This includes a cell line expressing a GRP94 T44A mutant, +/- treatment with the proteasome inhibitor, bortezomib, to highlight the efficient degradation of hyperglycosylated GRP94 in these Δ GRP94 RKO cells.

In Extended Data Fig. 5A, both mono- and hyperglycosylated GRP94 are present in the WT lane. This appears to conflict with the model that regulated N-glycosylation protects the nascent chain from hyperglycosylation. Clarification is needed regarding this apparent inconsistency. Additionally, it is unclear where Supplementary Fig. 5A is referenced or located—please verify its inclusion.

We apologize for the confusion. The experiment presented in Ext. Data Fig 5A (now Ext. Data Fig. 7a) shows a small amount of hyperglycosylated GRP94 in wild-type HEK293 cells transiently transfected with a wild-type Flag-GRP94 construct. In contrast, analysis of endogenous GRP94 in wild-type HEK293 and RKO cells (Ext. Data Fig. 7c,d) shows no detectable hyperglycosylation by immunoblotting. This difference is likely due to the higher levels of exogenous GRP94 in the transfected samples, which saturates the capacity of FKBP11 and CCDC134 (which are expressed at much lower levels) to fully suppress hyperglycosylation. Importantly, the mutants shown in Ext. Data Fig. 7a show a much stronger hyperglycosylation phenotype than the wild-type construct (all expressed at comparable levels), consistent with the model.

Minor points:

On the third line of page 3, 'CCCD₁₃₄' be corrected to 'CCDC₁₃₄? On the sixth line of page 4, 'OST-A' be corrected to 'OST-B'?

Thanks for these corrections.

The figure legends and labels could benefit from additional clarification to aid reader understanding. For example, in Extended Data Fig. 5B, does the "WT" entry under "GRP94-5N addback" refer to a GRP94 variant lacking all five facultative sequons but without any additional mutations in the pre-N segment? If so, it may be helpful to use a more descriptive term—such as "5N-only"—rather than "WT" to avoid confusion with the true wild-type GRP94.

We modified the labeling to avoid confusion.

The TRAP complex components (TRAP $\alpha/\beta/\gamma/\delta$) also appear to be affected by limited map quality. In particular, the density in certain regions does not support confident placement of side chains. If local map improvement is not feasible, the authors should consider simplifying the model by omitting side chains—at least in the poorly resolved regions—to prevent overinterpretation and reduce the risk of conveying potentially misleading structural information.

Density for the TRAP complex is indeed weak, particularly within the bilayer. As detailed in the Methods, the model (based on an AlphaFold prediction used in 8RJG) was placed into density as three separate rigid bodies (TRAP β,γ,δ membrane and cytosolic segments; TRAP α,β,δ luminal domains; and TRAP α TMD and cytosolic tail) and then fit using tightly restrained real-space refinement in Coot. Because the density does not support confident placement of sidechains, we restricted our analysis to its position relative to the rest of the translocon, and comparison to previously determined structures.

Reply to Reviewer 3

A serious hindrance in reviewing this manuscript was the scarceness of experimental details, and in some cases the inclusion of protocols that were not specific to this study. Figure 1 shows the cryo-EM structure of Sec61, TRAP, and OST-A together with a nascent GRP94 protein associated with these translocon components. The methods state that cells expressing a Flag-tagged version of FKBP11 were used as a source of microsomes, which were subjected to immunoprecipitation for the structural studies. The critical proteins for this study are not clearly resolved, but AlphaFold-derived structures of the three proteins were fit to a luminal density. What is the source of GRP94? I assume the cells are expressing a stalled version of this protein, but there is no mention of this in the methods.

We apologize for the confusion. Although we could not find protocols unrelated to this study, we have attempted to address the reviewer's concerns by providing additional experimental details throughout the main text, legends and methods sections (including for the new selective ribosome profiling experiments).

The reviewer is correct that the cryo-EM sample was prepared by affinity-purification of FKBP11-bound ribosomes from detergent solubilized microsomes obtained from a Flag-FKBP11 HEK293 cell line. As noted above, the GRP94 visualized in the structure is endogenous, native GRP94 (not stalled). Please see responses to Reviewers 1 and 2 for further discussion of the native GRP94 translocation intermediate seen in the cryo-EM density.

The point mutants support the model, and thus in spite of much of the structure being based on AlphaFold, it is very likely that this is the structure that supports suppression of the 4 non-native glycosylation sites. However, it remains unknown why this very intriguing and complex mechanism exists and under what conditions it would be used. The authors suggest that it could be used to detect misfolded GRP94, but misfolding would have to occur in the first ~40 amino acids of GRP94 to allow the suppressing interactions to occur. Is it possible that amino acid deprivation stress causes pausing and this stabilizes interaction of that "pseudo substrate" peptide?

Our model is that CCDC134 and FKBP11 limit hyperglycosylation of the facultative sites when GRP94 folds properly. Mutations would not necessarily be restricted to the pre-N domain. When something goes wrong—because of GRP94 mutation, loss or saturation of CCDC134, OST-A defects, exposure to ER stress etc.—these sites are no longer shielded, leading to hyperglycosylation.

Whether amino-acid deprivation stress can slow translation and stabilize the GRP94 inhibitory configuration is not known to us. We note that a cursory examination of the selective ribosome profiling data does not reveal obvious pausing, suggesting that under typical growth conditions (cell culture), pausing is not required to suppress GRP94 hyperglycosylation.

I was also curious as to how the native glycosylation site (N217) is modified, as it is downstream of the blocking peptide. Is this modified by STT3B? And if so, why are N445, N481, and N502 not glycosylated? When does disruption of the pre-peptide:CCDC134:STT3A complex occur? Sometime after N502 enters the lumen, I guess? I realize all of these questions are beyond the scope of this study, but maybe some discussion of these points would make the discussion a little more interesting.

Our working model is that N-domain folding is slow (relative to the M- and C-domain) due to its complex architecture and high contact order (see Ivankov et al., 2003). This would leave N217 accessible to OST-B (or OST-A), acting in trans, until it can fold against CCDC134. At this point, the now partially folded N-domain (i.e., what we see in the structure) contributes to a luminal vestibule in which the M-domain can attempt to fold while its facultative sites are shielded from OST-B (in trans). As demonstrated by our new selective ribosome profiling data (Fig. 1f), CCDC134 and OST-A engage early with GRP94, and remain bound until synthesis is essentially complete. We now clarify these points in the Discussion.

Reply to Reviewers 4 and 5

The EM map is based on an FKBP11 pulldown. While the occurrence of GRP94 is somewhat unexpected, numerous examples exist where chaperone pulldowns yielded exciting substrate-containing structures. Nevertheless, additional orthogonal evidence confirming that GRP94 is a component of the imaged complex would strengthen the work, given GRP94's central role and the relatively low resolution and conformational differences compared to mature GRP94. For example, intensity-based quantification from mass spectrometry could demonstrate the high abundance of GRP94. Alternatively, RNAseq (as used by the Keenan lab in McGilvray et al., 2020) could help establish GRP94's prevalence. An even more informative experiment—perhaps beyond the scope of this study—would be FKBP11 pulldown in a ΔGRP94 background to assess whether the complex still forms and how this affects structure and substrate binding.

As noted above, we performed IP/MS on FKBP11-bound ribosomes in an earlier study (DiGuilio et al., 2024, Fig 2C), and the result clearly confirms GRP94 as one of the most abundant components. The same study demonstrated the high abundance of GRP94 transcripts by RNA-seq. At the reviewer's suggestion, we also performed selective ribosome profiling on Flag-FKBP11 and Flag-CCDC134 samples (Fig. 1 and Ext. Data Fig 1). These data confirm our earlier results with FKBP11 and extend them to CCDC134.

The selective ribosome profiling experiment also informs on the reviewer's question about whether FKBP11/CCDC134 assembly is specific for GRP94. As shown in Fig. 1c,e, FKBP11 and CCDC134 also enriched hundreds of other transcripts (mostly proteins with long translocated segments), albeit much less robustly than GRP94. We speculate that FKBP11 and CCDC134 may have a broader function at the translocon—namely, to shield hydrophobic segments of unfolded nascent chain as they spool into the ER lumen. This point has been added to the Discussion.

The single-particle cryo-EM reconstruction of the ribosome-bound GRP94-translocon appears sound but is of limited resolution. While a global resolution of ~3.4 Å is reported, it is dominated by the ribosome and therefore provides limited insight into the translocon region. Local resolution appears substantially lower, especially for GRP94/FKBP11/CCDC134/KCP2/DC2 (~8–11 Å according to Extended Data Fig. 3C). This should be explicitly clarified in the text.

The re-processed data yield maps with global resolution of ~4.1 Å (Map 1, translocon-only) and ~3.1 Å (Map 2, whole particle focused on 6oS), and we now include model statistics separately for the translocon and the 6oS + translocon models (Ext. Data Table 1). As the reviewer correctly points out, however, these global statistics do not capture the considerable variation in local resolution. To aid the reader we provide additional examples of model-map fits (including comparison to the AF2 models) (Ext. Data Fig 5 and 6), and we are explicit about what maps are shown in all figures.

All model statistics (Extended Data Table 1) include the ribosome, which skews the interpretation of structural quality. Model statistics should be reported separately for the translocon, especially GRP94/FKBP11/CCDC134/KCP2/DC2. Model-to-map cross-validation metrics would also be highly relevant.

Thanks for the suggestion. We now report model statistics separately for the translocon and provide additional examples of model-to-map fits in these key regions.

It would be helpful if the authors included a ChimeraX session containing all reported maps and fitted models to allow a more detailed review.

This is a terrific suggestion—we have provided the requested ChimeraX session.

Due to the low resolution, AF2 modeling is crucial for assignment. While this is acceptable, the assignment is difficult to evaluate because only refined models are shown—which necessarily fit the maps. Pre- and post-refinement models superposed onto the maps should be included as supplementary material to illustrate the modeling process.

We apologize for this. Ext. Data Fig 5 now shows the final model-map fits with AF2 models superimposed for comparison.

Refinement at very low resolution is prone to overfitting. The authors wisely mention using "tightly restrained" real-space refinement, but the specific methodology and parameters used should be clearly defined in the Materials and Methods section.

The Methods section now includes additional details of the restraints applied during real-space refinement in Coot and Phenix.

The fit of the pre-N segment to cryo-EM density (Fig. 2c) is not very convincing regarding the precise register of the model, as sidechains, including bulky residues such as F76 and F78, are unresolved. Given the modest resolution, the detailed panels E–G also appear somewhat speculative.

As noted above, we re-processed the data to full resolution and performed a more extensive particle classification, yielding maps with improved density for most of the translocon components and GRP94. Notably, the pre-N domain sequence register is well-supported by sidechain densities for multiple residues (e.g., V27, V31, L35, R39, S42, R43, T44, V48, V49, R51, L61, I66, R70, F78) and by excellent agreement with the AlphaFold model (see Fig. 3c,d and Ext. Data 5b).

TRAP is visible in a previously published multipass-translocon map by McGilvray et al. (eLife, 2020), though it was not modeled. In this map of the multipass-translocon, the TRAP position also deviates substantially from later reconstructions of the same complex from other groups (e.g., Smalinskaitė et al., Nature 2022; Gemmer et al., Nature 2023). The authors should comment on whether specific solubilization protocols may possibly have altered TRAP's position in both cases.

Despite what is often very weak density, the position of the TRAP complex is clearly different in different structures. While it is possible that purifying samples in mild detergents like digitonin could affect TRAP position, this doesn't seem to be the dominant factor. For example, cryo-EM structures of detergent-solubilized RTCs show TRAP bound in its canonical position to the core translocon (8btk, 8rjc, 8rjd; open and closed Sec61 states), and in various displaced positions in the secretory (current structure; open Sec61)

and multipass translocons (6w6l, 7tut, others; closed Sec61). Likewise, cryo-ET structures determined in membranes show TRAP bound in its canonical position to the secretory translocon (8b6l; open Sec61), and in displaced position (disordered, but tethered via its C-terminal, ribosome-binding anchor) to the multipass translocon (closed Sec61) (Gemmer 2024). More work is needed to understand the factors that affect TRAP positioning. We currently favor the idea that the occupancy of accessory factors (e.g., BOS) and the translocated nascent chain are both important factors for TRAP positioning.

In any event, the main point we make in the current manuscript is that TRAP repositioning (for whatever reason) and packing of the partially folded GRP94 N-domain against CCDC134, creates a luminal vestibule which allows the M-domain to fold in a shielded environment that is protected from glycosylation in trans by OST-B. We have revised this section of the text to emphasize this point.

The assignment of KCP2 is particularly difficult to follow, as the AlphaFold2 multimer prediction primarily involves the very flexible N-terminal region of OSTC, making the tracing possibly challenging.

We apologize for not making this clearer. As shown in a new Ext. Data Fig. 5d, the AF2 prediction for KCP2-DC2 could be fit into the density essentially as a rigid body, with only minor adjustments to the flexible N-terminal linker of DC2.

Although AF2 modeling guided most of the structure assignments, the basis for FKBP11's ribosome binding is unclear. Given the resolution, assignment based solely on the EM data seems insufficient and should be better explained.

We apologize for not explaining this point more clearly. We previously showed that the C-terminal region of FKBP11 is required for ribosome binding (see Fig. 1, DiGuilio et al., 2024). This observation is fully consistent with the structure. We clarify this point in the main text, and also include a panel showing the model-map fit in this region (Ext. Data Fig. 6d).

Reply to Reviewers 1 and 2

The revisions have addressed the main concerns from the previous review, particularly with respect to cryo-EM map quality, model interpretation, and biochemical validation. The structural assignments are now presented more conservatively, and I find the revised manuscript suitable for publication.

I have only a few minor suggestions that could improve clarity in the final version:

Fig. 1f – The positional analysis shows distinct interaction profiles of GRP94 with CCDC134 and FKBP11. The GRP94–FKBP11 interaction reaches a high level rapidly before the synthesis of residue ~200, whereas the GRP94–CCDC134 interaction increases more gradually, peaking between residues ~350–400. Please consider clarifying your interpretation of these differing interaction kinetics and discussing the possible mechanistic basis for this temporal offset in recruitment.

We thank the reviewers for this suggestion. We have modified the Results and Discussion to more clearly describe the different interaction profiles of FKBP11 and CCDC134, and to clarify the proposed model.

Lines 114–116 and 159–160 – Consider referencing the relevant figures that show the densities for the LLO donor and the N-glycans, so that readers can more easily locate the supporting structural evidence.

The previous version of the manuscript did not show densities for the LLO donor and other glycans, but given this request we have added some examples to Extended Data Fig. 6b,c.

Reply to Reviewer 3

The authors have answered my questions in this revised version and I am satisfied by their responses. I support publication of this manuscript in Nature.

Reply to Reviewers 4 and 5

The additionally provided data greatly strengthen the revised manuscript and support the conclusions. The provided chimeraX network as well as the PDB report also help to better analyze strengths and weaknesses of the data. The cryo-EM data remain unexpected and exciting. However, the resolution of the translocon is mostly limited to the secondary structure level, which makes circumstantial evidence from alphaFold and ribosome profiling key to support the conclusions. The EM data still appear overinterpreted in some respects, and consistency of ribosome sequencing and image classification must be clarified.

Specific points:

- The authors nicely show significant enrichment of Grp94-coding mRNA in the pulldowns. Ultimately the RNA profiling should, however, support the assignment of cryo-EM density. Hence, an attempt should be made to directly correlate transcript abundance with cryo-EM image classification. The key question is: Does the RNAseq analysis at least approximately support the assignment of Grp94? According to the EM data (Ext. Fig. 2) > 10% of the total ~500k ribosome particles have GRP94 bound. Does this approximately match the RNA-seq quantification? In other words: is the ribosome profiling supporting the notion that 1/10 of all (presumably all elongating) ribosomes in the sample synthesize Grp94?

Yes, despite differences in the way the samples were prepared, there is good qualitative agreement between the two estimates. Of the 498,812 ribosome-translocon complexes (RTCs) selected by cryo-EM analysis (following initial classification and 3D refinement; Ext. Data Fig 2), 55,750 were included in the final reconstruction after focused classification—i.e., about 11.2% of the RTCs contain GRP94, FKBP11 and CCDC134. Selective ribosome profiling (via Flag-FKBP11) identified ~9,264 unique transcripts. GRP94 was the most abundant of these, representing about 4.6% of the total transcript abundance.

- This reviewer appreciates the effort to improve the resolution of the reconstruction, but the visualization of side chains in Fig. 1c/d remains unconvincing. The Q-scores in the PDB report (well below 0.5 for even for the pre-N segment of Grp94) do not support compelling fit of side chains. Exactly this, however, seems to be message of these panels, which should be avoided. What makes the assignment of density to Grp94 and the backbone model nevertheless solid in this reviewer's opinion is the very good agreement of the

AF2 backbone model and the EM density (Extended Data Fig. 5b). Hence, this reviewer suggests exchanging these two panels, i.e., making the comparison to AF a main figure as it is essential for the conclusions.

We apologize for the confusion. The Q-scores listed in the OneDep validation report are of limited value because they report the model fit to Map 2—the whole particle map after focused refinement on the 60S subunit. We selected this map for OneDep validation because the deposited coordinates (PDB ID 9NgJ) include the 60S subunit and the translocon. However, density for the translocon (and GRP94) is significantly stronger in the translocon-only map (Map 1). This is best appreciated by visual inspection of the maps, but is also illustrated using the ChimeraX Qscore tool (Pintille et al, 2020)—note the higher (>0.5) Q-scores for the GRP94 pre-N segment and STT3A (for example) in Map 1 compared with Map 2:

We are pleased that the reviewer is convinced by the agreement between the AF2 prediction and the EM density (Ext. Data Fig. 5b), and appreciate the suggestion. However, we prefer to keep Fig. 3c,d as it is currently shown to keep the AF2 comparisons in one place (Ext. Data Fig. 5b), and to avoid visually complicating one of the main figures. We think this is reasonable because the Fig. 3c,d already shows sidechain density for a number of residues in this region (e.g., V31, R39, S42, R43, T44, R51, and others).

- The modeling of the FKBP11 TM and cytosolic domains from the EM map are weak. The TM domain is not resolved at all (as opposed to the TMs is binds to according to the model). A short helix in the cytosol is resolved, yet tracing from lumen to cytosol is impossible. If the authors want to retain the ribosome-bound parts of FKBP11, the reviewer's suggestion is the following: (i) remove the TM domain from the PDB model, as it is not supported by the EM data. (ii) provide a difference map of a (previously published) FKBP11-free translocon density with their density to provide some EM-based support for this assignment. Again, the EM data alone are not sufficient for assignment, but comparative analysis may make it plausible.

Density for the FKBP11 TMD is indeed weak, but continuous density is visible across the membrane in low-pass filtered translocon-focused maps (e.g., Job588_Refine3D_6A.mrc in the ChimeraX session provided for review). This tube of density aligns with the cytosolic and luminal domains of FKBP11, supporting its assignment as the FKBP11 TMD. We have added this to Ext. Data Fig. 6f.

Point-by-point response

In response to the request by Referee #4 we deposited the translocon-only Map 1 and the corresponding translocon-only model to the EMDB (EMD-72945) and PDB (9YGY), respectively. Extended Data Table 1 and the Methods section are updated accordingly.

Referee #4 also suggested that we show “...EM density of a published ‘FKBP11-free’ ribosome-translocon as a reference to support the assignment of the C-tail”. Previously published “FKBP-free” RTC maps—including core, secretory and multipass translocons—do not show similar density in the region we assigned to the FKBP11 C-terminal helix. As requested, we now include one example of the absence of density in this region, from EMD-4316, in a new Extended Data Fig. 6g.